# CD206+ macrophages facilitate wound healing through interactions with *Gpnmb*hi fibroblasts

Azusa Honda[1,2], Hiroyuki Koike [ID][1,3], Teruyuki Dohi [ID][2], Eri Toyohara[2], Sumio Hayakawa[1], Kazuyuki Tobe[4], Ichiro Manabe[5], Rei Ogawa [ID][2] & Yumiko Oishi [ID][1,3][✉]

## Abstract

**Wound healing is a multifaceted and dynamic sequence of tissue repair and regeneration processes involving interrelated stages: inflammation, regeneration, and remodeling. Throughout these processes, macrophages change their phenotypes and interact with cells and extracellular components to facilitate healing. In particular, macrophages expressing the surface marker CD206 associate with inflammation resolution and tissue repair. However, how CD206+ macrophages contribute to these processes is insufficiently understood. Here, using a mouse model of CD206+ macrophage depletion and single-cell transcriptomics, we report that selective depletion of CD206+ macrophages results in modest but significant delays in wound healing, prolongs inflammation, and significantly reduces the number of *Gpnmb*hi fibroblasts in injured skin. Single-cell data suggest that CD206+ macrophages communicate with *Gpnmb*hi fibroblasts via multiple pathways. Notably, topical administration of PDGF-AA to wounds of CD206+ macrophage-depleted mice restores healing processes, identifying PDGF-A signaling from CD206+ macrophages to PDGFRA on fibroblasts as an important mechanism promoting wound healing. Collectively, these data demonstrate that CD206+ macrophages communicate with *Gpnmb*hi fibroblasts to activate their proliferation and extracellular matrix deposition in wound healing.**

Keywords Macrophages; Wound Healing; PDGFA; Hypertrophic Scar
Subject Categories Immunology; Signal Transduction; Stem Cells & Regenerative Medicine

## Introduction

Wound healing is a complex and dynamic tissue repair and regeneration process consisting of three overlapping phases: inflammation, proliferation, and remodeling (Baum and Arpey, 2005; Singer and Clark, 1999). During these processes, multiple cell types interact in time- and space-specific manners to repair and regenerate damaged skin. Neutrophils are the first cells to infiltrate the injury site and initiate the inflammatory phase by producing high amounts of reactive oxygen species, proteases, and pro-inflammatory cytokines (Wang, 2018). Next, there is an accumulation of large numbers of inflammatory monocytes and macrophages (Krzyszczyk et al, 2018). Macrophages clean the wound area by phagocytosing any bacteria and tissue debris so that the site is ready for the proliferative phase of tissue regeneration. In the proliferative phase, cells residing in the damaged tissue, including fibroblasts and endothelial cells, proliferate vigorously and generate provisional granulation tissue, which is characterized by vascularized extracellular matrix (ECM) containing macrophages. Epithelial cells then migrate into the granulation tissue to close the wound by epithelialization (Gurtner et al, 2008).

Macrophages not only clean the wound early for repair, but are also critically involved in all subsequent phases of proliferation and regeneration (Mirza et al, 2009; Shook et al, 2016). They accumulate at the site of injury and shift from pro-inflammatory to pro-resolution phenotypes during the wound healing process. During the early stages of wound healing, inflammatory monocytes were recruited in a CCR2-dependent manner and differentiated into Ly6Chi macrophages to clean the wound. Next, Ly6CloF4/80hi macrophages become predominant, with the majority of these macrophages expressing the surface marker CD206, which is encoded by *Mrc1*. Some of these macrophages proliferate within the skin (Davies et al, 2013). While macrophage depletion at only the very early stage (days 0–1) reduced fibroblast numbers on day 7 post-injury without significantly affecting other healing processes (Shook et al, 2016), depletion of macrophages that spans day 2 and beyond has been shown to impair re-epithelialization, vascularization, and fibroblast proliferation (Mirza et al, 2009; Shook et al, 2016, Rodero et al, 2010). Likewise, macrophage depletion during only the mid-stage (days 3–7) severely impaired wound healing (Lucas et al, 2010). Because the transition from Ly6Chi to Ly6Clo macrophages occurs on days 2–3 (Mirza et al, 2009; Shook et al, 2016), these findings indicate that Ly6Clo macrophages during the mid-stage play a crucial role in skin regeneration after injury. It is also very likely that Ly6Chi macrophages are crucial for preparing

[1]Department of Biochemistry & Molecular Biology, Nippon Medical School, 1-1-5 Sendagi, Bunkyo-ku, Tokyo 113-8602, Japan. [2]Department of Plastic, Reconstructive and Aesthetic Surgery, Nippon Medical School, 1-1-5 Sendagi, Bunkyo-ku, Tokyo 113-8602, Japan. [3]Department of Medical Biochemistry, Graduate School of Medical and Dental Sciences, Institute of Science Tokyo, Bunkyo-ku, Tokyo, Japan. [4]First Department of Internal Medicine, University of Toyama, 2630 Sugitani, Toyama-shi, Toyama 930-0194, Japan. [5]Department of Systems Medicine, Graduate School of Medicine, Chiba University, 1-8-1 Inohana, Chuo-ku, Chiba-shi, Chiba 260-8670, Japan. [✉]E-mail: oishi.yumiko@tmd.ac.jp

wounded tissues for repair and regeneration, as shown in injury responses in other tissues (Graubardt et al, 2017).

A previous study showed that the predominant macrophage population during the mid-stage of skin wound healing (3–5 days post-injury) expresses CD206 and can be further divided into CD301b-positive and CD301b-negative subpopulations. Among these, both the number of CD206$^+$CD301b$^+$ macrophages within the wound and their relative proportion within the CD206$^+$ population increase during the mid-stage period (Shook et al, 2016). Notably, depletion of CD301b$^+$ macrophages 3 days after injury impaired accumulation of fibroblasts, re-epithelialization and vascularization in a skin punch injury model (Shook et al, 2016), indicating that CD206$^+$CD301b$^+$ macrophages play an important role in skin regeneration. However, the precise function of CD206$^+$ macrophages in skin wound healing has not been assessed in vivo, and the cellular communication that occurs between CD206$^+$ macrophages and other cell types present in wounds is not well understood.

Fibroblasts are essential for skin wound healing. After skin damage, fibroblasts are activated and proliferate within the wound. These activated fibroblasts pivotally contribute to wound healing, partly by depositing ECM and also by differentiating into myofibroblasts that contract the wound (Martin, 1997). Recent single-cell-based and lineage-tracing analyses have shown that fibroblasts are a much more heterogeneous cell population than previously thought (Abbasi et al, 2020; Guerrero-Juarez et al, 2019; Phan et al, 2021). Certain fibroblast populations localize to specific regions within mouse skin and have distinct abilities to synthesize ECM to perform different functions during wound healing (Buechler et al, 2021b). For example, fibroblasts derived from the lower dermis have a higher capacity to synthesize collagen and elastin than fibroblasts derived from the upper dermis (Driskell et al, 2013). Furthermore, progeny of lower dermal fibroblasts contribute early, while progeny of upper fibroblasts migrate into the wound later and may contribute to hair follicle regeneration (Phan et al, 2021). In addition, expression of the engrailed-1 gene (En1) has been shown to mark wound fibroblasts producing ECM (Mascharak et al, 2021). Indeed, fibroblasts in normal dorsal mouse skin do not express En1; however, after skin injury, En1-expressing fibroblasts are the predominant fibroblasts present in wounds. Depletion of such fibroblasts reduces ECM deposition and delays wound closure (Rinkevich et al, 2015). With the functional and phenotypic heterogeneity among fibroblasts, it is expected that fibroblast subpopulations differentially interact with the different macrophage subpopulations in wounds to contribute to various processes that occur during skin wound repair. Given that macrophages and fibroblasts influence the development of fibrosis and scarring, crosstalk between macrophages and fibroblasts may become a therapeutic target to prevent excessive scarring. In this study, we have determined the fibroblast subsets that interact with CD206$^+$ macrophages and how these macrophages instruct the fibroblast subsets to repair the wound.

# Results

## Depletion of CD206$^+$ macrophages delays wound healing

Previous studies have shown that the depletion of myeloid cells impairs skin wound healing and can affect collagen production and scaring of the wounds (Joost et al, 2018; Lucas et al, 2010; Wang et al, 2019). However, the specific function of CD206$^+$ macrophages in this process is insufficiently understood. Therefore, we decided to investigate the role of CD206$^+$ macrophages in skin wound healing using Mrc1-DTR transgenic mice (Nawaz et al, 2022) in which expression of the diphtheria toxin receptor gene is driven by the Mrc1 (CD206) promoter. Peritoneal administration of diphtheria toxin to these mice specifically depletes CD206$^+$ macrophages throughout the body. To evaluate the effects of diphtheria toxin-mediated depletion of CD206$^+$ macrophages on skin wound healing, we used the standard splinted wound model (Dunn et al, 2013). For this model, full-thickness, excisional wounds (5 mm in diameter) were made on the backs of mice, and ring-shaped silicone splints were applied to the skin to prevent wound contraction caused by the panniculus carnosus and to more closely mimic human wound healing. Because Mrc1-DTR is established in the C57/BL6 background, we used wild-type mice with the same genetic background as controls (Nawaz et al, 2017; Nawaz et al, 2022). Diphtheria toxin was administered to wild-type and Mrc1-DTR mice on days 0 and 1 after wounding and every other day until 2 days before analysis (Fig. 1A). Diphtheria toxin administration did not affect the gross appearance of the mice.

To assess the effects of diphtheria toxin-mediated CD206$^+$ macrophage depletion on wound healing, we analyzed the cells in wounded tissues 5 days after wounding. As reported previously (Shook et al, 2016), large numbers of CD45$^+$CD11b$^+$F4/80$^+$ macrophages were present in the wound. Among these, the CD206$^+$ cells constituted the major population (~85%; Fig. 1B gating strategy is shown in Fig. EV1A–C). The amount of CD206$^+$ macrophages was increased after wounding and was most abundant at 5 days post wounding (Fig. EV1D). However, in the Mrc1-DTR mice, diphtheria toxin administration reduced F4/80$^+$CD206$^+$ cells present in the wound to ~40% of the level in control wound (Fig. 1C). The number of CD206$^+$ macrophages in wounds 5 days post-injury was also evaluated by immunostaining and confirmed that their numbers were reduced in the dermis of the Mrc1-DTR mice (Fig. 1D,E).

We assessed wound healing by determining the ratio of the wound area to the area of the splint. At day 3 post-injury, there was no difference in wound area between diphtheria toxin-treated WT and Mrc1-DTR mice. However, 5 days after injury, the wound areas of Mrc1-DTR mice were significantly larger than those of the control mice (71% vs. 61% of the splint area, $P = 0.0072$; Fig. 1F,G). This delayed wound closure persisted until 11 days after injury. These results indicated that the depletion of CD206$^+$ macrophages impaired wound healing, and that the delay in healing became apparent 5 days after injury (Fig. 1F,G). On the 5th day, the re-epithelialization process has been delayed and mildly increased F4/80$^+$ macrophages were observed in the wounds of the Mrc1-DTR mice (Fig. 1H,I).

To further investigate the effect of depletion of CD206$^+$ macrophages from the wound, we performed bulk RNA sequencing (RNA-seq) of the wound on day 5. Gene set enrichment analysis (GSEA) (Fang et al, 2023; Subramanian et al, 2005) of the Molecular Signatures Database (MSigDB) Hallmark gene sets (Castanza et al, 2023) showed significant upregulation of the gene sets related to inflammation, such as IL6 JAK STAT3 signaling and inflammatory response, and downregulation of the gene sets related to cell proliferation and epidermal development in Mrc1-DTR wounds compared to the control (Fig. 1J), supporting the

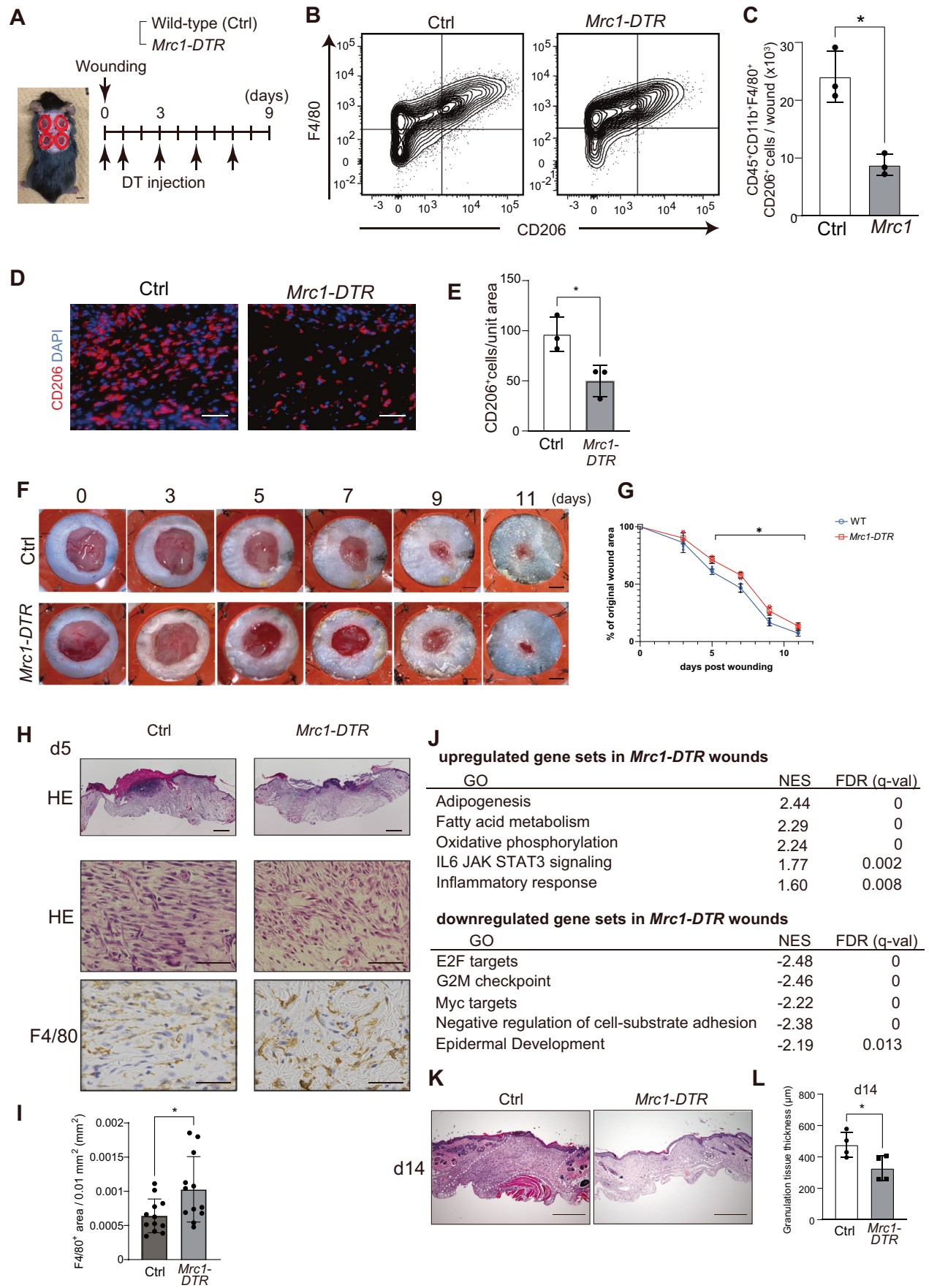

**Figure 1. Depletion of CD206⁺ macrophages impairs wound healing.**

(A) Schematic diagram of the diphtheria toxin (DT) injection and wounding model used to analyze mouse skin repair. An image of the gross appearance of the mice (scale bars, 5 mm) is shown on the left. (B, C) Cells were collected from the wound beds of *Mrc1-DTR* and control mice on day 5 after injury and subjected to flow cytometry analysis. Representative plots of CD45⁺CD11b⁺ myeloid cells are shown (B). The number of CD45⁺CD11b⁺F4/80⁺CD206⁺ macrophages per wound was counted by flow cytometry and compared between groups (C). Data are shown as means ± SD. *$P = 0.0289$ by unpaired two-tailed Student's *t* test ($n = 5$ mice/group, biological replicates). Wound regions were defined as areas with full-thickness skin loss in the 5 mm diameter circular excision. (D, E) Representative images of wounds from *Mrc1-DTR* and control mice immunostained for CD206. Scale bars, 100 μm (D). The numbers of CD206⁺ cells/0.1 mm² wound area were compared between groups (E). Data are shown as means ± SD. *$P = 0.0250$ by unpaired two-tailed Student's *t* test, ($n = 5$ mice/group, biological replicates). (F) Representative images of wounds obtained from *Mrc1-DTR* and control mice. Scale bars, 1 mm. (G) Wound healing curve showing the percentage of wound closure over the time period. Data are shown as means ± SD. *$P = 0.0072$ (day 5), $P = 0.0121$ (day 7), $P = 0.0307$ (day 9), $P = 0.0020$ (day 11) by two-way ANOVA followed by Tukey's post hoc test ($n = 4$ mice/group, biological replicates). (H) Representative histologic images of wound tissues that were harvested 5 days after wounding, then stained with hematoxylin and eosin (HE) and immunostained for F4/80. Scale bars, 500 μm for the low-magnitude images, and 100 μm for the high-magnitude images. The white dotted line indicates the original wound edge. (I) F4/80 positive area/0.01-mm² wound area were compared between mouse groups. Data are shown as means ± SD. *$P = 0.0202$ by unpaired two-tailed *t* test after applying logit transformation ($n = 12$ areas/group, biological replicates). (J) Top 5 Molecular Signatures Database (MSigDB) hallmark gene sets that were enriched in *Mrc1-DTR* wounds compared to the control wounds in GSEA are shown. NES, normalized enrichment score. (K) Representative histological images of granulation tissue harvested and stained with HE 14 days after wounding. The white dotted line indicates the healed wound area, corresponding to the original wound boundary. Scale bars, 500 μm. (L) The thickness of granulation tissue was measured on day 14 after wounding. Data are shown as means ± SD. *$P = 0.0403$ by unpaired two-tailed Student's *t* test ($n = 5$ mice/group, biological replicates). Source data are available online for this figure.

prolonged inflammation and delayed re-epithelialization process. Furthermore, 14 days after injury, the granulation tissue was significantly thinner in *Mrc1-DTR* mice than in WT mice, even after complete wound closure (Fig. 1K,L). Collectively, these results suggest that CD206⁺ macrophages are required for skin regeneration with thick granulation tissue formation.

## Single-cell analysis reveals diverse cell populations in wound tissue

Our findings that re-epithelialization process and wound closure were significantly attenuated by CD206⁺ macrophage depletion led us to analyze the intercellular communication occurring at the wound site. We collected whole live cells from wound tissues 5 days after wounding and analyzed them by single-cell RNA sequencing (scRNA-seq) (Fig. 2A). After quality control measures were taken, we had 12,058 cells from *Mrc1-DTR* mouse wounds and 11,031 cells from control mouse wounds. We then combined the two datasets and identified 12 clusters by unsupervised clustering (Fig. 2B). Cell-type identities were attributed to the clusters using differentially expressed genes and marker gene expression (Dataset EV1). Representative differentially expressed genes and their expression levels in each cluster are shown in the heatmap and the uniform manifold approximation and projection (UMAP) plots in Fig. 2C,D. Cluster C0 was enriched for macrophage markers, such as *Lyz2*, *Cd68*, and *Fcgr1*, and Cluster C1 was enriched for epithelial markers, including *Lgals7*, *Krt14*, and *Krt17*. Clusters C2 and C3 were classified as fibroblasts because they abundantly expressed ECM genes, including collagen genes, such as *Col1a2* and *Col3a1*. Cluster C2 cells also expressed anti-fibrotic and ECM degradation-signature genes, such as *Ctsk*. Cluster C4 cells were classified as dendritic cells (DCs) based on the expression of genes involved in inflammation and elastic tissue integrity, such as *Ccr7*, *Ccl22* and *Tbc1d4*. The remaining clusters (C5–11) were classified as neutrophils, natural killer (NK) cells, pericytes, Langerhans cells, endothelial cells, mast cells, and myocytes, respectively.

When we examined the effects of CD206⁺ cell reduction on the cell populations present in wounds, we found that in control tissues, immune cells, including macrophages, dendritic cells, neutrophils, and NK cells, accounted for ~30% of the cells

analyzed. However, in wounded *Mrc1-DTR* tissues, the proportion of immune cells was much larger (78%), and there were fewer fibroblasts and epithelial cells (Fig. 2E,F). Flow cytometry analysis confirmed that the CD31⁻Epcam⁻CD26⁺ fibroblast population was significantly decreased and the number of CD45⁺ immune cells was significantly increased in the wound bed of *Mrc1-DTR* mice (Fig. EV1E,F). These findings are also consistent with the histological findings showing the persistent accumulation of F4/80⁺ macrophages and the thinner granulation tissue in *Mrc1-DTR* mice (Fig. 1H). In addition, we analyzed the populations of immune cells in the scRNA-seq datasets. Notably, the proportions of neutrophil and macrophage were increased among immune cell population in the wound of *Mrc1-DTR* mice (Fig. 2G).

We then analyzed the macrophage subpopulations in *Mrc1-DTR* wounds. Unsupervised clustering of the macrophage populations identified 6 distinct subclusters: *C1qa⁺*, *Vcan⁺*, *Ccl7⁺*, *Cd9⁺*, *Klf2⁺*, and *Ifit1⁺* macrophages (Fig. 3A,B). *C1qa⁺* (cluster 0) populations expressed high levels of complement C1q subcomponent subunit genes, such as *C1qa* and *C1qc* as well as *Pltp*. *Vcan⁺* populations (cluster 1) expressed high levels of *Plac8* and *Il1b*. *Ccl7*+ (cluster 2) populations also expressed high level of *Ccl2*. *Ifit1⁺* populations expressed high levels of interferon-related genes, such as *Ifit1*, *Ifit2* and *Ifit3* (Fig. 3C,D). *Mrc1* is expressed at relatively high levels in *C1qa⁺* (cluster 0) and *Ccl7⁺* (cluster 2) populations (Fig. 3D).

Differential gene expression between macrophages in *Mrc1-DTR* wounds and macrophages in control mice was analyzed by Gene set enrichment analysis (GSEA) (Fang et al, 2023; Subramanian et al, 2005). GSEA of the Molecular Signatures Database (MSigDB) Hallmark gene sets (Castanza et al, 2023) showed significant upregulation of the gene sets related to inflammatory activation, such as inflammatory response, IL6 JAK STAT3 signaling, TNFα signaling via NF-κB in *Mrc1-DTR* wounds (Fig. 3E). Next, the differential composition of macrophages was analyzed. Notably, *C1qa⁺* macrophages (cluster 0) were markedly reduced in *Mrc1-DTR* wounds. *C1qa⁺* macrophages were dominant populations, accounting for 84% of the macrophages in the wounds of control mice, whereas they were only 11% of the macrophages in *Mrc1-DTR* wounds. In contrast, the *Vcan⁺*, *Ccl7⁺*, and *Cd9⁺* (clusters 1–3) subpopulations accounted for less than 10% in the control wounds, whereas these populations increased to

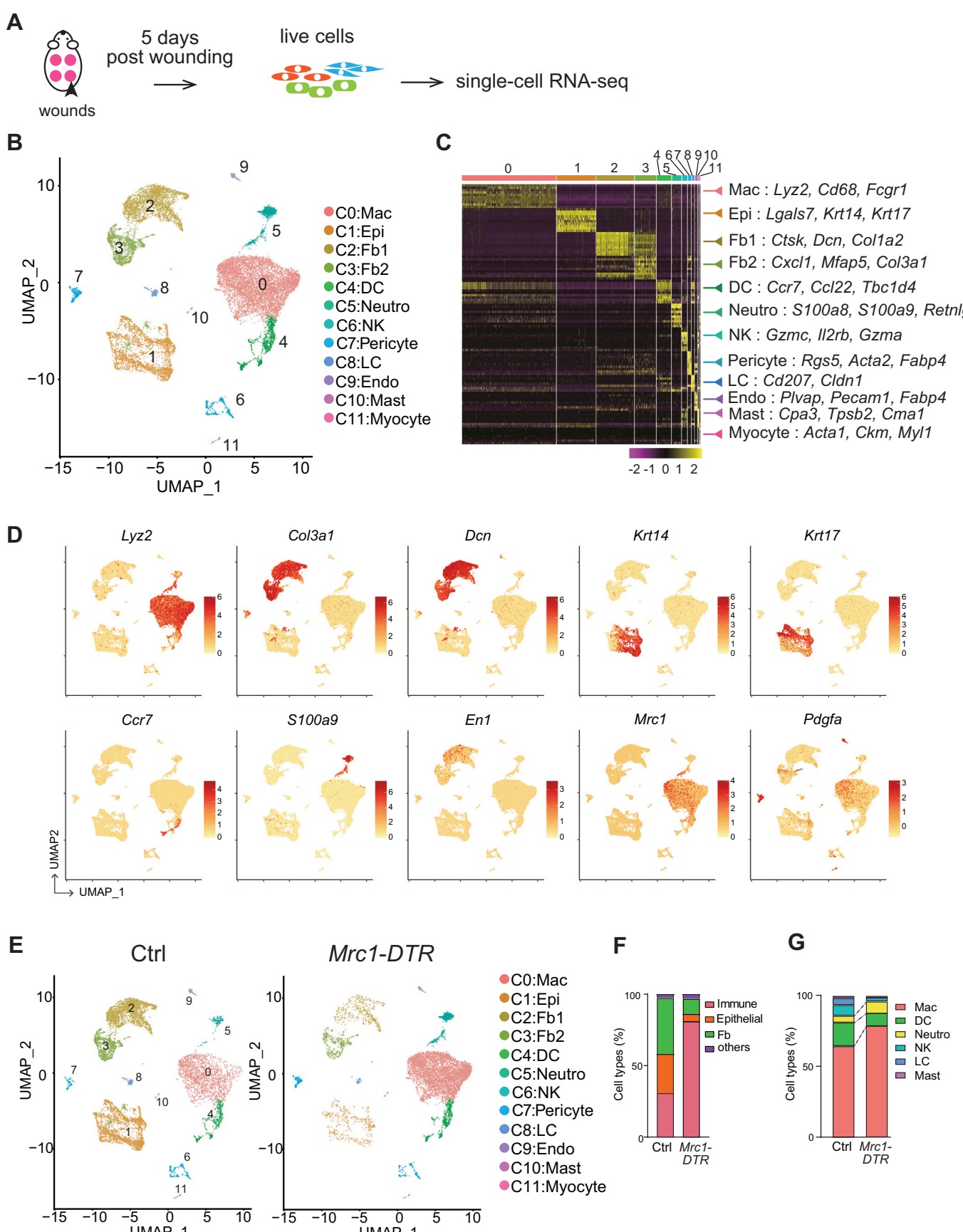

**Figure 2.  scRNA-seq reveals distinct skin wound components in CD206-depleted mice.**

(A) Schematic illustration detailing the single-cell isolation strategy. (B) UMAP plot showing cellular heterogeneity. Twelve distinct clusters of wound cells were identified and color-coded. The general identity of each cell cluster is indicated on the right. (C) Heatmap showing the top 10 genes enriched in each cluster. Selected genes from each cluster are listed on the right. Mac macrophages, Epi epithelial cells, Fb fibroblasts, DC dendritic cells, Neutro neutrophils, LC Langerhans cells, Endo vascular endothelial cells. (D) Feature plots illustrating the expression distribution of the indicated genes. The expression levels in each cell were color-coded and overlaid onto UMAP plots, as shown. (E) UMAP plots showing the cellular heterogeneity of wound bed cells from control and *Mrc1-DTR* mice. (F) Bar chart showing the percentages of immune cells, fibroblasts (Fb) and epithelial cells in the wound beds of control and *Mrc1-DTR* mice. (G) Bar chart showing the proportions of cell types among the immune cells.

34%, 29%, and 21%, respectively, in the *Mrc1-DTR* wounds (Fig. 3F,G).

To gain insights into the functions of macrophage subpopulations, we analyzed the enrichment of gene sets among the differentially expressed genes in each subpopulation. Our analysis revealed that the macrophage subpopulations exhibited distinct patterns of functional term enrichment (Fig. EV2; Dataset EV2). $C1qa^+$ (cluster 0) cells were characterized by expression of the genes related to extracellular matrix organization and metabolism and angiogenesis, suggesting they are involved in tissue remodeling and wound healing. The other $Mrc1^{hi}$ subpopulation, $Ccl7^+$ (cluster 2) was characterized by gene sets related to lipoprotein clearance. The most abundant subpopulations in *Mrc1-DTR* wounds, $Vcan^+$ (cluster 1), and $Cd9^+$ (cluster 3), expressed gene sets related to inflammation. $Cd9^+$ subpopulation was also characterized by expression of glucose-metabolism related genes, suggesting it included pro-inflammatory activated macrophages. $Klf2^+$ (cluster 4) and $Ifit1^+$ (cluster 5) subpopulations were characterized by high enrichment in TNF-α signaling and interferon signaling, respectively. The fractions of these subpopulations appeared to be less affected by *Mrc1-DTR*-mediated depletion. Collectively, these findings support the notion that *Mrc1-DTR*-mediated depletion reduced macrophage subpopulations contributing to wound repair and increased those characterized by pro-inflammatory activation.

## Depletion of CD206⁺ macrophages decreases the number of cells from specific fibroblast subsets in wounds

Since *Mrc1-DTR* wounds were characterized by delayed and thin granulation tissue formation, we next focused on analyzing the fibroblast populations in our scRNA-seq datasets. Three subclusters exhibiting distinct transcriptomes—Fb1 ($Gpnmb^{hi}$), Fb2 ($Plac8^{hi}$), and Fb3 ($Crabp1^{hi}$)—were identified in the wound tissues (Fig. 4A–C). Recent scRNA-seq analyses have revealed heterogeneity among wound fibroblasts and have also identified subpopulations that show distinct spatial distribution in wounds (Guerrero-Juarez et al, 2019; Lim et al, 2018). *Crabp1* expression marks fibroblasts in the upper dermis (i.e., upper wound fibroblasts), while *Plac8* and *Mest* expression mark fibroblasts localized to the deep, reticular layer (lower wound fibroblasts) (Joost et al, 2020; Phan et al, 2021). Therefore, in our experiments, Fb2 ($Plac8^{hi}$) cells are mainly lower wound fibroblasts and Fb3 ($Crabp1^{hi}$) cells are mainly upper wound fibroblasts. Fb1 ($Gpnmb^{hi}$) cells can be found in both upper and lower wound areas.

Among the three fibroblast subpopulations we identified, the loss of CD206⁺ cells most profoundly reduced the proportion of $Gpnmb^{hi}$ fibroblasts, with the percentage of $Gpnmb^{hi}$ cells among the total fibroblasts decreasing from 60% in WT to 25% in *Mrc1-DTR* wounds (Fig. 4D). To gain insights into the functional features of $Gpnmb^{hi}$ fibroblasts, we performed gene set enrichment analysis on the differentially expressed genes in $Gpnmb^{hi}$ cells compared to the rest of the fibroblasts in WT mice using GSEApy (Fang et al, 2023). We found that sets of genes related to extracellular matrix organization and extracellular structure organization were enriched in $Gpnmb^{hi}$ fibroblast populations, suggesting that these cells contribute to ECM deposition (Fig. 4E).

$Gpnmb^{hi}$ cells expressed higher levels of *En1* and were enriched in $En1^{hi}$ cells (Fig. 4C,F). *En1*-expressing fibroblasts have been shown to promote ECM deposition and scar formation after injury, further supporting the notion that $Gpnmb^{hi}$ fibroblasts contribute to ECM deposition (Mascharak et al, 2021). Taken together, these results suggest that $Gpnmb^{hi}$ fibroblasts are involved in granulation tissue formation by promoting ECM deposition and organization, and that having decreased numbers of $Gpnmb^{hi}$ fibroblasts delays healing and wound closure in CD206⁺ macrophage-depleted mice.

Because $En1^{hi}$ fibroblasts were enriched in the $Gpnmb^{hi}$ fibroblast population, we analyzed EN1 expression in fibroblasts by immunostaining in day 5 wounds. The localization of EN-positive fibroblasts was confirmed using CD26 as an upper dermis (papillary dermis) fibroblast marker and FAP as a marker in the lower dermis (reticular dermis) layer. In uninjured skin, EN-positive fibroblasts were scattered throughout the dermal layer (Rinkevich et al, 2015), whereas in the wound bed, and they localized to the deep dermal layer where CD206⁺ macrophages are accumulated (Fig. 5A,B). The number of EN1⁺ fibroblasts per 0.01 mm² of wound area was significantly reduced in CD206⁺ macrophage-depleted wounds compared to control wounds (Fig. 5B,C). In addition, the percentage of EN1⁺ cells co-stained with Ki67 was significantly decreased in the wounds of CD206⁺ macrophage-depleted mice compared to those of control mice (Fig. 5D,E). Collagenous connective tissue fiber deposition was also attenuated in the wounds of CD206⁺ macrophage-depleted mice (Fig. 5F), indicating that CD206⁺ macrophages affect the proliferation of EN1⁺ fibroblasts and collagenous connective tissue fiber deposition.

To further assess effects of CD206⁺ macrophage depletion on fibroblasts, we performed GSEA (Subramanian et al, 2005) of differential gene expression in the fibroblast subpopulations (Table EV1). The depletion of CD206⁺ macrophages led to enrichment in the cholesterol homeostasis and mTORC1 signaling gene sets across all three fibroblast populations, suggesting CD206⁺ depletion not only affected fibroblast population compositions but also their cellular metabolism.

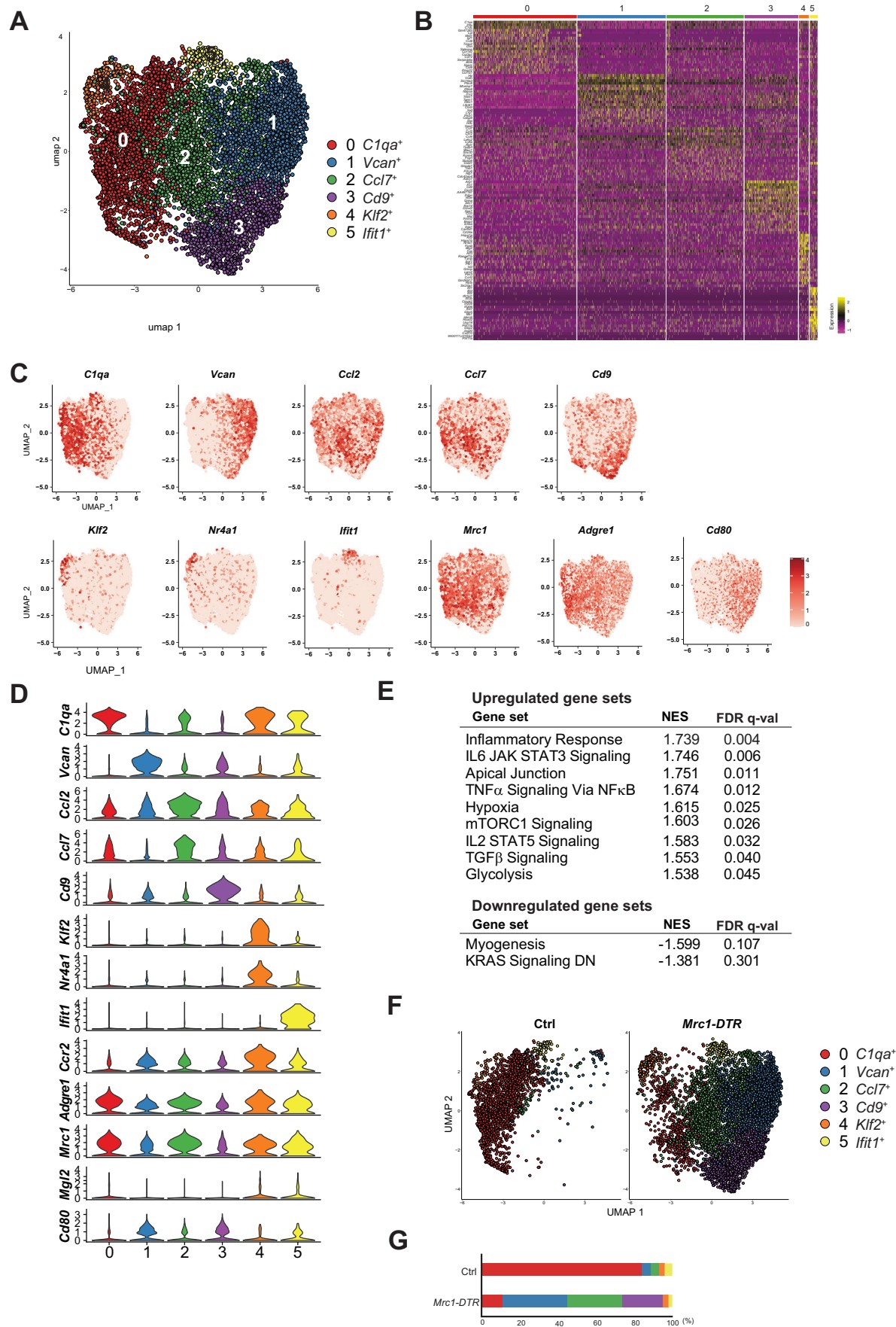

**Figure 3. scRNA-seq analysis of macrophage populations in control and *Mrc1-DTR* wounds.**

(A) UMAP plots of macrophages from day 5 wound beds. scRNA-seq data of wound beds obtained from control and *Mrc1-DTR* mice were combined and analyzed. (B) Heatmap of the top 10 genes enriched in each macrophage subcluster. (C) Feature plots of expression distribution for selected cluster-specific genes (*C1qa*, *Vcan*, *Ccl2*, *Ccl7*, *Cd9*, *Klf2*, *Nr4a1*, *Ifit1*), *Mrc1*, *Adgre1* and *Cd80*. Gene expression levels for each cell were color-coded and overlaid onto UMAP plots. (D) Stacked violin plots showing the expression of the indicated genes in the macrophage subclusters (*n* = 1). (E) Molecular Signatures Database (MSigDB) hallmark gene sets that were enriched in macrophages of *Mrc1-DTR* wounds compared to the control wounds in GSEA are shown. NES, normalized enrichment score. (F) UMAP plots of macrophages from day 5 wound beds from control and *Mrc1-DTR* mice. (G) Bar graph showing the percentages of macrophages in control and *Mrc1-DTR* wounds 5 days post-injury.

## CD206+ macrophages promote wound healing via PDGF-A

The observation that depleting CD206+ macrophages impaired *Gpnmb*hi fibroblast proliferation prompted us to further examine *Gpnmb*hi fibroblast-CD206+ macrophage communication mediated by ligand–receptor interactions. For this analysis, we combined *Plac8*hi and *Crabp1*hi fibroblasts to form a *Gpnmb*lo fibroblast population. Then potential interactions between the fibroblast and macrophage populations were analyzed using LIANA+ (Dimitrov et al, 2024; Dimitrov et al, 2022) (Fig. EV3). Among the top 20 ligand–receptor pairs we narrowed candidate pairs by the following criteria: (1) *C1qa*+ macrophages showed stronger interactions with *Gpnmb*hi fibroblasts, as compared with the subpopulations that were increased in *Mrc1-DTR* (*Vcan*+, *Ccl7*+, *Cd9*+) particularly *Vcan*+, which was the most abundant subpopulation in *Mrc1-DTR*, and (2) *C1qa*+ macrophages showed stronger interactions with *Gpnmb*hi fibroblasts, as compared to *Gpnmb*lo fibroblasts. We identified *Pdgfa-Pdgfra*, *Igf1-Itgav_Itgb3*, *Pdgfra-Pdgfrb*. Among the receptors of these pairs, the expression level of *Pdgfra* was higher in *Gpnmb*hi cells than *Gpnmb*lo cells (Fig. 4C,F), so we decided to focus our studies on PDGF-A signaling.

Although PDGF-B is well known to promote wound healing (Pierce et al, 1991; Pierce et al, 1989), much less is known about PDGF-A in wound healing. Our cell-cell interaction analysis showed that PDGFA-PDGFRA was ranked the highest among the PDGF signaling pathways for *C1qa*+ macrophage-*Gpnmb*hi fibroblast interactions (Fig. 6A). As expected, PDGF-A signals colocalized with CD206 signals in immunofluorescence staining of wound bed tissues obtained from control mice 5 days after wounding. This is consistent with the scRNA-seq result showing that the macrophage population (in particular, the *Mrc1*hi macrophage population) expresses *Pdgfa* mRNA, although *Pdgfa* mRNA is expressed in other cell types such as Langerhans cells, endothelial cells and epithelial cells (Fig. 2D). In contrast, PDGF-A signals, particularly the proportion of PDGF-A-expressing CD206+ cells, was significantly reduced in wound tissue obtained from CD206+ macrophage-depleted mice, further supporting the idea that CD206+ macrophages are the major producers of PDGF-A in the wound (Fig. 6B,C). Immunostaining also showed that EN1+ fibroblasts expressed PDGFRA and that a higher proportion of EN+ cells expressed PDGFRA than EN- cells (Fig. 6D,E).

To evaluate the effects of PDGF-AA (PDGF-A homodimer) on fibroblast functions, we cultured primary fibroblasts derived from neonatal (P1) mouse skin and treated them with mouse recombinant PDGF-AA. This exposure to exogenous PDGF-AA significantly increased *En1* mRNA expression (Fig. 6F). It also increased the expression of the collagen genes, *Col1a1* and *Col3a1*. These data suggest that CD206+ macrophage induces *En1* and collagen gene expression in fibroblasts.

Next, we examined the effects of PDGF-AA treatment on skin wound healing in vivo. For these experiments, recombinant mouse PDGF-AA was topically administered to the wound bed surface of *Mrc1-DTR* mice 24 h after wounding. Our results showed that PDGF-AA treatment accelerated wound closure, as indicated by the smaller wound areas on days 5 and 9 after wounding (Fig. 6G,H). These data demonstrate that PDGF-AA supplementation at least partially ameliorates the delayed wound healing of *Mrc1-DTR* mice. Moreover, we performed immunofluorescence staining and found that the proportion of fibroblasts co-expressing EN1 and Ki67 was increased by PDGF-AA treatment, indicating that PDGF-AA augmented the proliferation of EN1+ fibroblasts (Fig. 6I,J). Collectively, our findings demonstrate that CD206+ macrophages interact with a specific subset of fibroblasts that express *Gpnmb* and *En1* to promote wound closure and healing mediated by PDGF-AA in mice.

Delayed wound healing and prolonged inflammation lead to hypertrophic scars and keloids in human (Xu et al, 2020). Therefore, we further investigated whether the CD206+ macrophage-dependent wound healing observed in mice also applies to human keloids. Immunohistochemical analysis of human keloid tissues revealed the simultaneous presence of both CD206+ macrophages and EN1+ fibroblasts within the lesions (Fig. EV4). These results suggest that, similar to mice, CD206+ macrophages and EN1+ fibroblasts also play roles in wound healing in humans.

## Discussion

The results of this study demonstrate the essential role of CD206+ macrophages in the control of a specific fibroblast subset in the process of skin wound healing and repair. First, we found that CD206+ macrophages interact with a subtype of fibroblasts expressing *Gpnmb* to enhance their proliferation, which is pivotal for wound healing and repair. Second, we identified PDGF-A-PDGFRA signaling as a key pathway that mediates communication between CD206+ macrophages and *Gpnmb*hi fibroblasts. Supporting this notion, topical administration of PDGF-AA to skin wounds of CD206+ macrophage-depleted mice was shown to successfully restore the healing process.

Recent lineage-tracing and scRNA-seq analyses have revealed that skin fibroblasts are a highly heterogeneous population (Correa-Gallegos et al, 2023). After skin injury, spatiotemporally distinct fibroblast subpopulations emerge in skin wounds and appear to have distinct functions. Several markers for these fibroblast subpopulations have been identified. Among them, *En1* is induced in fibroblasts in response to injury, and *En1*-expressing fibroblasts are the predominant cell type contributing to ECM

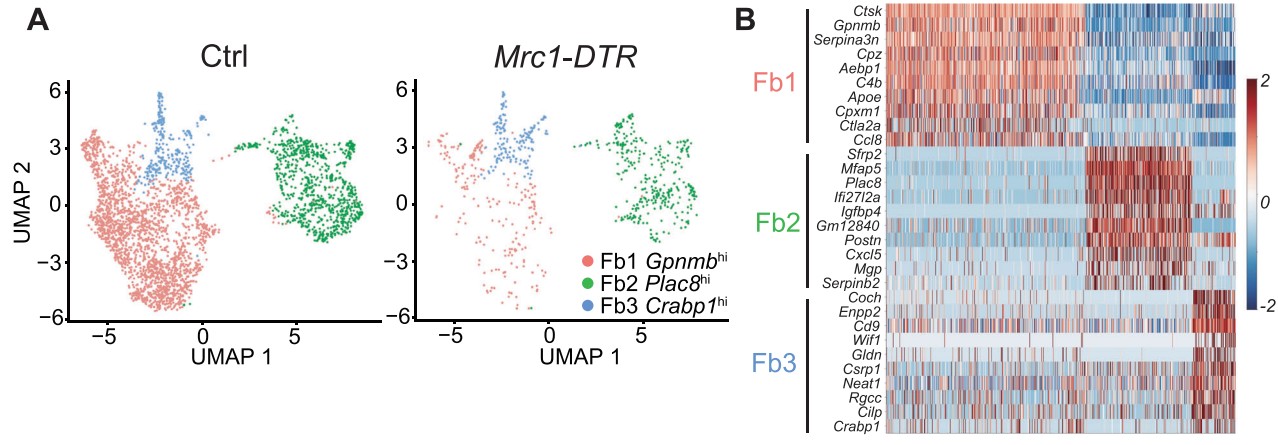

**A** Ctrl *Mrc1-DTR*

Fb1 *Gpnmb*hi
Fb2 *Plac8*hi
Fb3 *Crabp1*hi

**B**

Fb1, Fb2, Fb3 gene heatmap

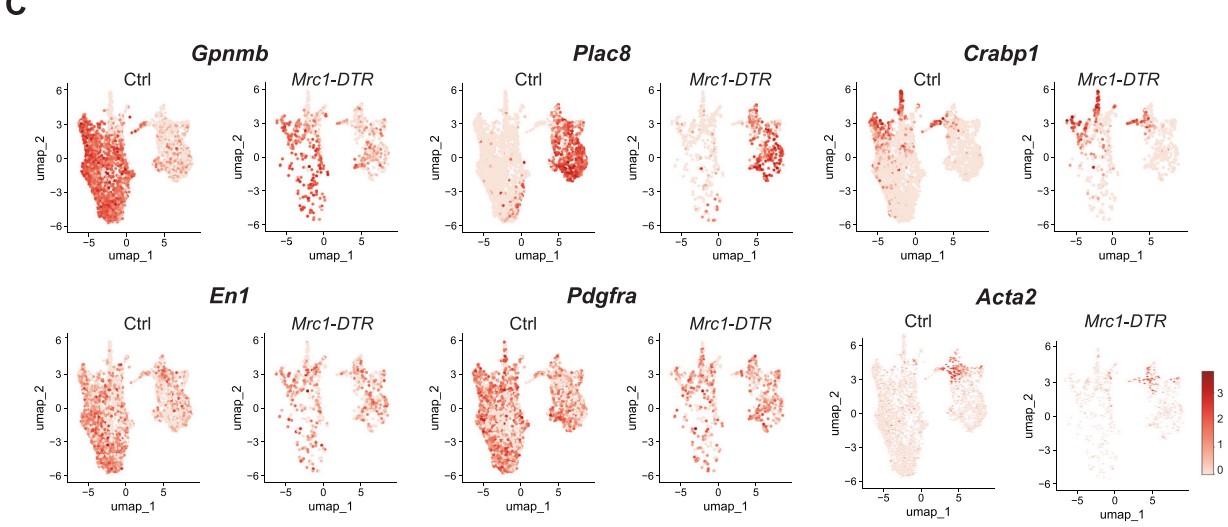

**C**

*Gpnmb* *Plac8* *Crabp1*

*En1* *Pdgfra* *Acta2*

**D**

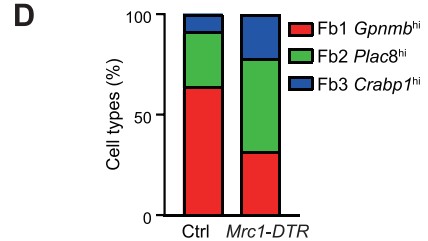

Fb1 *Gpnmb*hi
Fb2 *Plac8*hi
Fb3 *Crabp1*hi

**E**

### Upregulated gene sets in *Gpnmb1*hi subpopulations

| Gene set | NES | FDR q-val |
|---|---|---|
| Regulation of T cell proliferation | 1.934 | 0.061 |
| Lipid transport | 1.875 | 0.069 |
| Negative regulation of Endothelial cell migration | 1.799 | 0.148 |
| Regulation of ERK1 and ERK2 cascade | 1.738 | 0.199 |
| Extracellular matrix oraganization | 1.449 | 0.194 |
| Extracellular structure organization | 1.425 | 0.235 |

### Downregulated gene sets in *Gpnmb1*hi subpopulations

| Gene set | NES | FDR q-val |
|---|---|---|
| Peptide biosynthetic process | -2.887 | 0 |
| Canonical Wnt signaling pathway | -1.769 | 0.061 |
| Actin filament organization | -1.764 | 0.060 |
| Muscle organ development | -1.739 | 0.075 |

**F**

*Gpnmb*
*Plac8*
*Crabp1*
*En1*
*Pdgfra*

Fb1 Fb2 Fb3

◀ **Figure 4.  Depletion of CD206⁺ macrophages alters the size and composition of the fibroblast population in wounds.**

(A) UMAP plots of fibroblasts from the wound beds of control and *Mrc1-DTR* mice 5 days after wounding. (B) Heatmap of the top 10 genes enriched in each fibroblast subcluster. (C) Feature plots showing the expression distributions of selected cluster-specific genes (*Gpnmb*, *Plac8* and *Crabp1*), as well as *En1*, *Pdgfra* and *Acta2*. The expression levels for each cell were color-coded and overlaid onto UMAP plots, as shown. (D) Bar graph showing the percentages of the fibroblast subclusters present (out of the total fibroblasts) in wounds from control and *Mrc-DTR* mice. (E) Enrichr GO biological process gene sets that were enriched in *Gpnmb*ʰⁱ fibroblasts (FDR < 0.25) compared to the other fibroblast populations are shown. NES, normalized enrichment score. (F) Stacked violin plots showing the expressions of *Gpnmb*, *Plac8*, *Crabp1*, *En1*, and *Pdgfra* in the fibroblast subclusters (*n* = 1).

deposition in granulation tissues (Rinkevich et al, 2015). Skin injury induces *En1* expression in fibroblasts localized in the deep, reticular dermis in response to mechanotransduced signals, partly via the YAP pathway (Mascharak et al, 2021), although *En1*-inducing mechanisms in other fibroblast subpopulations are not clear. Our study demonstrates that PDGF-AA increases *En1* expression in fibroblasts in vitro (Fig. 5D) and increases the number of EN1⁺ fibroblasts in wounds. In addition, we found that *En1*ʰⁱ cells were enriched in the *Gpnmb*ʰⁱ fibroblast subset, which was reduced by CD206⁺ macrophage depletion, further supporting the notion that CD206⁺ macrophages regulate the proliferation of *En1*ʰⁱ fibroblasts. Taken together, although PDGF-A can be produced by endothelial cells and fibroblasts as well as by macrophages, our results suggest that PDGF-A produced by CD206⁺ macrophages is at least partially involved in the *En1* expression and the proliferation of EN1⁺ fibroblasts. Our results also indicate that, in addition to mechanotransduction, macrophage-fibroblast communication is important for the regulation of ECM deposition and granulation tissue formation in skin wounds, supporting the previous study showing that the CD301b⁺ macrophages, a subset of CD206⁺ macrophages, promote the myofibroblast proliferation (Shook et al, 2018).

Clinically, delayed wound repair results in prolonged inflammation, leading to the formation of abnormal scarring, such as hypertrophic scars and keloids, which are fibroproliferative skin disorders (Finnerty et al, 2016). Thus far, no efficient, molecularly and cellularly-targeted treatment or prevention methods have been established to inhibit the formation of hypertrophic scars or keloids (Lee and Jang, 2018). Although CD206⁺ macrophages are indispensable for wound healing and tissue repair, sustained CD206⁺ macrophage infiltration may prolong the proliferation and activation of *En1*⁺ fibroblasts and contribute to hypertrophic scar formation. In support of this notion, our immunohistochemical staining of samples from the earlobe region of keloid patients showed the accumulation of CD206⁺ macrophages and the presence of EN1⁺ fibroblasts (Fig. EV5). Mascharak et al, reported that depletion of *En1*⁺ fibroblasts suppresses scar formation and promotes regeneration in mice (Mascharak et al, 2021). Thus, time- and space-dependent regulation of CD206⁺ macrophage function is likely to be an effective therapeutic target to accelerate healing without pathological scar formation.

Monomeric PDGF proteins form four types of homodimers (PDGF-AA, PDGF-BB, PDGF-CC, and PDGF-DD) as well as one heterodimer (PDGF-AB) that bind to PDGF receptors (PDGFRA and PDGFRB) with different affinity (Fredriksson et al, 2004). In general, PDGF-A and -C bind to PDGFRA and PDGF-B and -D bind to PDGFRB in vivo. PDGF-BB is the best-characterized member of the PDGF family in skin wounds (Oefner et al, 1992). It regulates many cellular processes after skin injury, including

inflammatory cell recruitment, fibroblast migration, (Andrae et al, 2008) collagen deposition (Zubair and Ahmad, 2019), and granulation tissue formation (Grotendorst et al, 1985). Furthermore, recombinant human PDGF-BB protein gel (becaplermin) has been approved by the US Food and Drug Administration (FDA) and is used clinically for the treatment of diabetic neurogenic foot ulcers (Smiell et al, 1999). Compared to PDGF-BB, much less is known about the mechanistic role of PDGF-AA in wound healing, although several lines of evidence point to its pivotal involvement in such processes. For instance, hydrogel containing PDGF-AA has been shown to modestly promote wound closure (Fan et al, 2021). Wu et al, also showed that PDGF-AA mediates the pro-healing effect of transplanted adipose-derived stem cells (Wu et al, 2019). In addition to the exogenous effects of PDGF-AA, endogenous PDGF-A expressed in fibroblasts and endothelial cells expressing the cellular senescence marker p16^INK4a has been shown to promote myofibroblast activation and wound contraction in skin wounds created by a 6 mm punch without a silicone ring support (Demaria et al, 2014). Moreover, systemic deletion of PDGFRA, which has a higher affinity for PDGF-A, resulted in decreased accumulation of fibroblasts and ECM deposition in sponge discs implanted in dorsal subcutaneous tissues, without affecting the number of infiltrated macrophages (Horikawa et al, 2015). The PDGF-AA-PDGFRA axis also induces angiogenesis, which is essential for wound repair (Laschke et al, 2006). Adding to these previous studies, our results clearly demonstrate that PDGF-A is important for ECM-producing fibroblast proliferation and re-epithelialization after skin injury. In particular, using the silicone ring-supported wound model that prevents wound contraction by myofibroblasts, we were able to show that PDGF-AA increased EN1⁺ fibroblast proliferation. Collectively, the PDGF-AA-PDGFRA axis appears to control crucial wound healing processes, including fibroblast proliferation, myofibroblast activation, and angiogenesis. Our findings strongly suggest that PDGF-A produced by CD206ʰⁱ macrophages promotes *En1*⁺ fibroblast proliferation. However, PDGF-A is also produced by non-myeloid cells, including fibroblasts. Thus, future studies are necessary to elucidate the spatiotemporally distinct roles of PDGF-AA-PDGFRA signaling in cell-cell communication during wound healing and scar formation.

This study has several limitations that should be acknowledged. First, the use of *Mrc1-DTR* transgenic mice combined with the silicon ring-placement wound model may not fully represent the wound healing in humans. Second, the mechanisms by which CD206⁺ macrophages regulate fibroblast proliferation and ECM deposition remain incompletely understood. Our results demonstrate that PDGF-A signaling represents one of the key pathways mediating communication between CD206⁺ macrophages and *Gpnmb*ʰⁱ fibroblasts. However, as suggested by our ligand–receptor

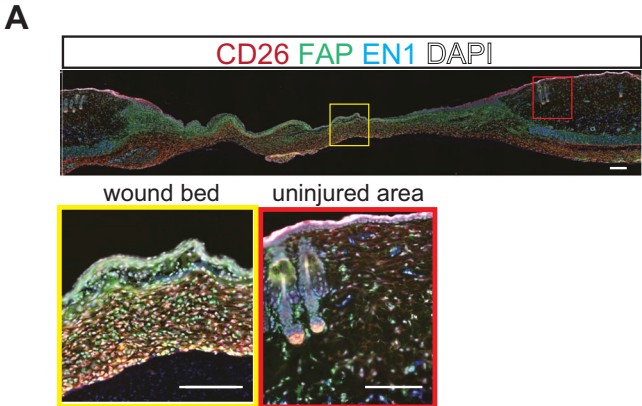

**A**

CD26 FAP EN1 DAPI

wound bed    uninjured area

**B**

Ctrl                    *Mrc1-DTR*

CD206 EN1 DAPI

**C**

EN1⁺ cell number/area (0.01 mm²)

**

Ctrl    *Mrc1 -DTR*

**D**

Ctrl                    *Mrc1-DTR*

Ki67 EN1 DAPI

**E**

DAPI-stained cells/ 0.01 mm² wound area (%)

■ EN1⁻
■ EN1⁺Ki67⁻
■ EN1⁺Ki67⁺

Ctrl    *Mrc1 -DTR*

**F**

Ctrl                    *Mrc1-DTR*

**Figure 5. Depletion of CD206+ macrophages reduces the number of EN1+ fibroblasts in the wound bed.**

(A) Representative images of wounds from control mice immunostained for CD26 (red, upper dermis fibroblast marker), FAP (green, lower dermis fibroblast marker), EN1 (blue) and DAPI (White). Scale bars, 200 μm (n = 3). (B, C) Representative images of wounds from Mrc1-DTR and control mice immunostained for CD206 and EN1. Low-magnification images (top panel) and high-magnification images (bottom panel) are shown. Scale bars, 50 μm (B). The numbers of EN+ cells/0.01-mm² wound area were compared between the two mouse groups (C). Data are shown as means ± SD. *P = 0.0040 by unpaired two-tailed Student's t test (n = 3 mice/group, biological replicates). (D, E) Representative images of wounds from Mrc1-DTR and control mice immunostained for Ki67 and EN1. Low-magnification (top panels) and high-magnification images (bottom panels) are shown. White arrowheads indicate EN1+Ki67+ cells. Scale bars, 50 μm (D). The percentages of EN1−Ki67− cells, EN1+Ki67− cells and EN1+Ki67+ cells out of the total DAPI-stained cells/0.01-mm² wound area were compared between mouse groups (E). Data are shown as means ± SD. *P = 0.0045 (EN1−), 0.3201 (EN1+ Ki67−), 0.0076 (EN1+ Ki67+) by two-tailed t test after applying logit transformation (n = 4 mice/group, biological replicates). (F) Representative Masson's trichrome-stained images of wounds from Mrc1-DTR and control mice. Scale bars, 200 μm (n = 3 mice/group, biological replicates). Source data are available online for this figure.

analysis (Fig. EV3), multiple additional signaling pathways are likely involved in this complex cellular interaction during wound healing. In addition, a recent study showed that macrophages enhance the YAP/TAZ pathway to promote myofibroblast contraction (Ezzo et al, 2024). Since YAP/TAZ signaling was also shown to be important for organ fibrosis (He et al, 2022), the effect of CD206+ macrophages on YAP/TAZ signaling in fibroblasts will need to be addressed. Third, although the differences were statistically significant, the magnitude of inhibition of healing processes in Mrc1-DTR mice was modest. One potential reason for the modest effect is that the administration of diphtheria toxin only partially depleted CD206+ cells. It is also possible that the remaining CD206+ macrophages may alter their functions, particularly at later time points, since wound tissue microenvironment changes over the course of inflammation to healing. As such, we need to further analyze spatiotemporal changes in cell subpopulations and their dynamic interactions throughout the repair process. In addition, although macrophage populations are the major cell type expressing Mrc1, Mrc1 is also expressed on other immune cells, such as dendritic cells and certain subsets of neutrophils (Fig. 2D). Potential off-target effects of diphtheria toxin on other CD206-expressing cells might influence the wound healing process.

In summary, our findings indicate that the communication between CD206+ macrophages and Gpnmbhi fibroblasts partly via PDGF-A-PDGFRA is pivotal for skin wound healing. Macrophages and fibroblasts are likely to interact with each other in various physiological and pathological processes, though previous studies have mainly focused on fibrosis and tumor growth (Buechler et al, 2021a). For example, interfering macrophages have been shown to modulate fibrosis in several tissues, including skin, lung, heart, and kidney. Recent advances in single-cell technologies have identified a variety of subpopulations of macrophages and fibroblasts and have suggested potential interactions. The finding that CD206+ macrophages communicate with Gpnmbhi fibroblasts provides mechanistic evidence for subpopulation-specific interactions in wound healing. Further elucidation of dynamic communications among macrophage and fibroblast subpopulations would not only reveal the responses and functions of cellular communities but also identify the mechanisms that can be targeted for therapeutic strategies for wound healing. Overall, we expect the results of the present study to facilitate the elucidation of the biological function of macrophage-fibroblast communication not only in skin but also in other tissues. Because fibrosis and remodeling are the hallmarks of organ dysfunction in chronic noncommunicable diseases and aging, elucidation of cellular communications between

macrophages and fibroblasts may also lead to identification of novel therapeutic and diagnostic targets in age-associated diseases.

# Methods

**Reagents and tools table**

| Reagent/resource | Reference or source | Identifier or catalog number |
|---|---|---|
| **Experimental models** | | |
| Mrc1-DTR (M. musculus) | Nawaz et al, 2017 | N/A |
| C57BL/6J (M. musculus) | Sankyo Labo Service | N/A |
| Human earlobe keloid | This study | N/A |
| **Recombinant DNA** | | |
| Recombinant Mouse PDGF-AA | Bioplegend | 776308 |
| **Antibodies** | | |
| CD45-Brilliant Violet 421 | Biolegend | 103133 |
| CD11b-PE/Cy7 | BD Pharmingen | 552850 |
| F4/80-PE | BD Pharmingen | 565410 |
| CD206-PE/Dazzle594 | Biolegend | 141731 |
| CD31-PE/Cy7 | BD Pharmingen | 561410 |
| CD26-FITC | Biolegend | 137805 |
| Anti-CD206 | Bio-Rad | MCA2235GA |
| Anti-engrailed-1 | Thermo Fisher Scientific | PA5-14149 |
| Anti-CD26 | R&D Systems | AF954 |
| Anti-FAP | R&D Systems | MAB9727 |
| Anti-PDGF-A | Santa Cruz Biotechnology | SC-9974 |
| Alexa Fluor 488 conjugated anti-rabbit | Invitrogen | A-11070 |
| Alexa Fluor 546 conjugated anti-rat | Invitrogen | A-11081 |
| Alexa Fluor 647 conjugated anti-mouse | Jackson ImmunoResearch | 115-607-187 |
| Anti-mannose receptor | Abcam | ab64693 |
| Anti-engrailed-1 | Bioss | BS-11744R |
| **Oligonucleotides and other sequence-based reagents** | | |
| PCR primer | This study | Table EV1 |

| Reagent/resource | Reference or source | Identifier or catalog number |
|---|---|---|
| **Chemicals, enzymes, and other reagents** | | |
| Carprofen | Zoetis | 170742 |
| Depilatory cream | Kracie | 4901417840417 |
| Isoflurane | Viatris | 901036504 |
| Diphtheria toxin | Sigma | D0564-1MG |
| Liberase TL | Roche | 5401020001 |
| RPMI | FUJIFILM | 4548995066268 |
| 7AAD | BD Pharmingen | 555816 |
| FBS | Cytiva | SH30910.3 |
| D-PBS | Nacalai | 14249-24 |
| Chromium Single Cell 3′ Reagent Kit v3 | 10x Genomics | PN-1000092 |
| 7.5% BSA in PBS | Thermo Fisher | 1520037 |
| NEBNext Ultra RNA Library Prep kit | New England Biolab | E7530 |
| Ethanol | Nacalai | 14713-95 |
| Xylene | Nacalai | 36611-45 |
| Tissue Tek OCT compound | Sakura Finetek | 4583 |
| Nucleospin RNA | Macherey-Nagel | 740955.50 |
| TaqMan™ Fast Advanced Master Mix for qPCR | Appliedbiosystems | 01225284 |
| ReverTra Ace qPCR RT Master Mix with genomic DNA (gDNA) Remover | TOYOBO | FSQ-201 |
| DMEM | Nacalai | 08458-16 |
| Penicillin/Streptomycin | Nacalai | 09367-34 |
| L-Ascorbic Acid Phosphate | FUJIFILM | 4987481395756 |
| PDGF-AA | BioLegend | 776302 |
| **Software** | | |
| FACS DIVA software | https://www.bdbiosciences.com/en-eu/products/software/instrument-software/bd-facsdiva-software | |
| FlowJo Software v10 | https://www.flowjo.com/flowjo/download | |
| Cell Ranger pipeline (v 6.0.0) | https://www.10xgenomics.com/support/software/cell-ranger/latest/release-notes/cr-release-notes | |
| Seurat v4 and v5 | https://satijalab.org/seurat/ | |
| SCALA | https://github.com/PavlopoulosLab/SCALA | |
| decoupleR | https://saezlab.github.io/decoupleR/ | |
| LIANA+ | https://liana-py.readthedocs.io/en/latest/ | |
| CellChat | http://www.cellchat.org | |
| STAR | https://github.com/alexdobin/STAR | |
| HOMER | https://docs.seqera.io/multiqc/modules/homer | |
| GSEA | https://www.gsea-msigdb.org/gsea/index.jsp | |
| DESeq2 | https://bioconductor.org/packages/release/bioc/html/DESeq2.html | |

| Reagent/resource | Reference or source | Identifier or catalog number |
|---|---|---|
| ImageJ | https://imagej.net/ij/ | |
| Prism software | https://www.graphpad.com | |
| **Other** | | |
| BD FACS Aria III | BD Biosciences | |
| Chromium Controller | 10x Genomics | |
| Agilent Bioanalyzer 2100 | Agilent | |
| Illumina Hiseq 2500 | Illumina | |
| Novaseq | Illumina | |
| BZ-X810 microscope | Keyence | |
| QuantStudio 5 | Thermo Fisher Scientific | |

## Animals

*Mrc1-DTR* mice were the same as those used previously (Nawaz et al, 2017). Male mice that were 7–9 weeks old were used in each experimental group. All mice were maintained in our institution's animal facility, with a 12 h/12 h light-dark cycle and free access to food and water. The standard splinted wound model was done as previously described (Dunn et al, 2013). In brief, the surgical site was prepared by removing fur from the base of the neck to 3 cm further down the back and between the two shoulder blades with clippers. Depilatory cream was applied, and the cream and remaining fur were removed with a wet gauze. A biopsy punch was used to create four full-thickness excisional wounds of 5 mm in diameter on the backs of each mouse under isoflurane anesthesia. A silicone ring was placed around each wound with adhesive and sutured to prevent healing by contraction. For moist wound healing, wounds were covered with a transparent occlusive dressing, which was changed every other day under isoflurane anesthesia. Carprofen (5 mg/kg) was administered once daily via subcutaneous injection for post operative pain relief. Each wound site was photographed on a fixed day, and the relative wound area was calculated as wound area divided by the area of the splint hole. All experimental procedures were conducted according to the protocol approved by the President of Nippon Medical School after being reviewed by the Nippon Medical School Animal Care and Use Committee (Approval No. H30-12). We adhered to the relevant guidelines and regulations concerning the management and handling of experimental animals. This study is reported in accordance with the ARRIVE guidelines (https://arriveguidelines.org).

## Diphtheria toxin injection

Diphtheria toxin (DT; Sigma, St. Louis, MO) was dissolved in water to the desired concentration for injection. To deplete CD206-expressing cells, 10 µl of 40 ng/µl DT was intraperitoneally injected into *Mrc1-DTR* mice and WT mice every other day. Wound tissue was harvested 2 days after the last injection.

## Preparation of single-cell suspensions

For the scRNA-seq and flow cytometry analyses, mouse wound bed and back skin were captured using a 10 mm biopsy punch. The recovered tissue was then minced with scissors and digested with

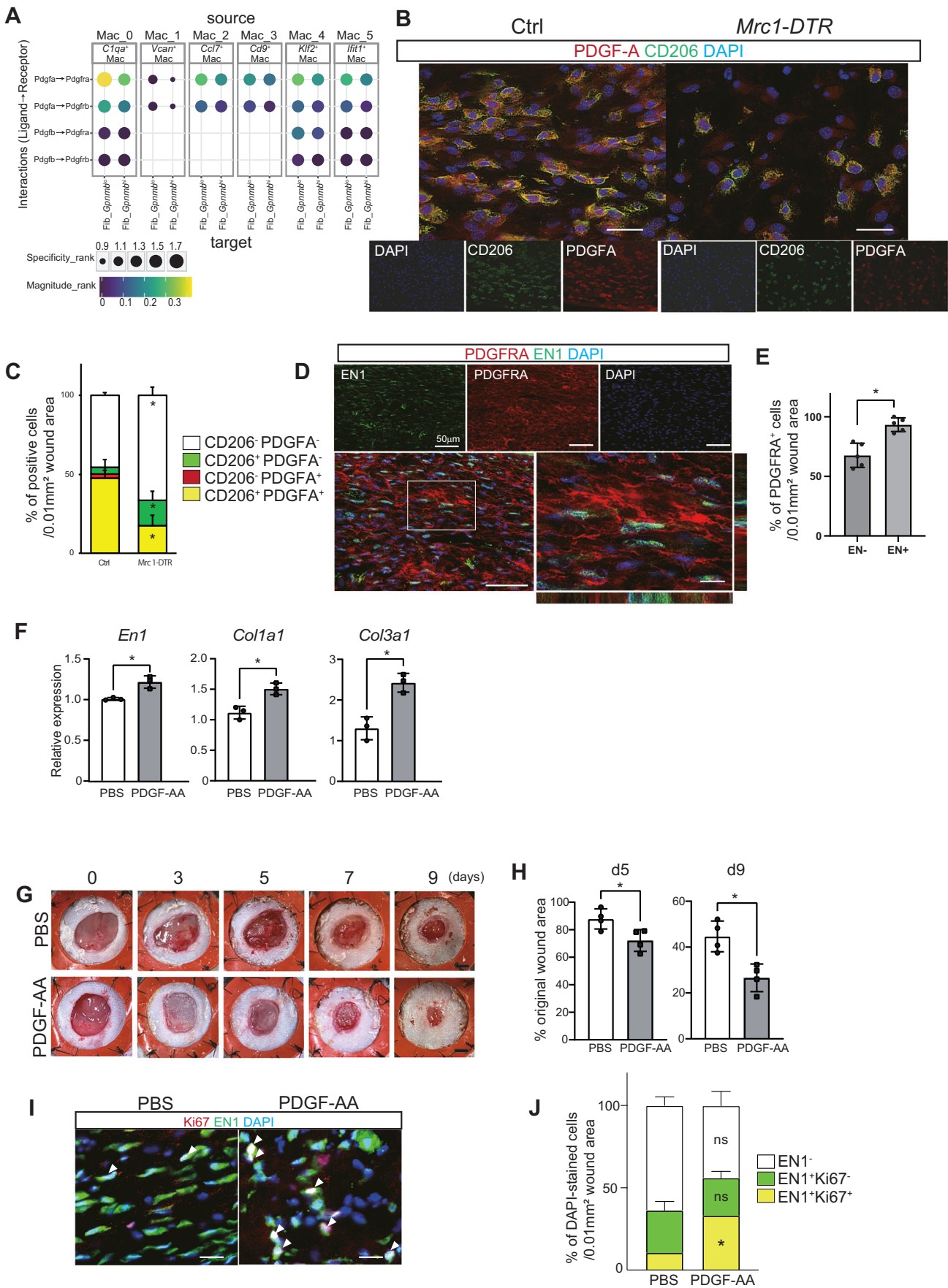

◀ **Figure 6. CD206$^{hi}$ macrophage-derived PDGF-A increases En1$^+$ fibroblast proliferation to promote wound healing.**

(A) Dot plots of the ligand–receptor interactions related to PDGF signaling in macrophage subpopulations and *Gpnmb*$^{hi}$/*Gpnmb*$^{lo}$ fibroblasts. Macrophage subpopulations that express ligands and fibroblast subpopulations that express receptors are shown on the *x* axis. Ligand and cognate receptor combinations are shown on the *y* axis. Circle size denotes *P* value (permutation test); color denotes average ligand and receptor expression levels in interacting subpopulations. (B) Representative images of wounds from *Mrc1-DTR* and control mice 5 days post-injury. Wound bed tissue samples were immunostained for PDGF-A (red) and CD206 (green). Scale bar, 50 μm. (C) The percentages of CD206$^-$, CD206$^+$PDGFA$^-$, CD206$^-$PDGFA$^+$ and CD206$^+$PDGFA$^+$ cells out of the total DAPI-positive cells/0.01-mm$^2$ wound area were compared between mouse groups. Data are expressed as means ± SD. *P* = 0.0003 (CD206$^-$ PDGFA$^-$), 0.0407 (CD206$^+$ PDGFA$^-$), 0.1351 (CD206$^-$ PDGFA$^+$), 0.0012 (CD206$^+$ PDGFA$^+$) by unpaired two-tailed *t* test after applying logit transformation (*n* = 4 mice/group, biological replicates). (D) Representative images of wounds from control mice 5 days post-injury. Wound bed tissue samples were immunostained for PDGFRA (red) and EN1 (green). Magnified images of orthogonal views of 2.4 μm z-stack images are shown on the right side. Scale bars, 50 μm (left) and 10 μm (right). (E) Comparison of the proportion of EN-negative cells co-expressing PDGFA with the proportion of EN-positive cells co-expressing PDGFA. Data are expressed as means ± SD. *P* = 0.0012 by unpaired two-tailed Student's *t* test (*n* = 5 mice/group, biological replicates). Fluorescence images with an IgG-negative primary antibody control are available in Fig. EV5. (F) The mRNA expressions of *En1*, *Col1a1*, and *Col3a1* in primary mouse fibroblasts treated with PDGF-AA (40 ng/mL) or PBS (control), as determined by qPCR, are shown. Expression levels were first normalized to those of *Gapdh* and then to the levels of control cells. Data are expressed as means ± SD. *P* = 0.0236 (*En1*), 0.0087 (*Col1a1*), 0.0059 (*Col3a1*) by unpaired two-tailed Student's *t* test. (*n* = 3/group, biological replicates). (G) Representative images of wounds of *Mrc1-DTR* mice treated with PDGF-AA or PBS. Scale bars, 1 mm. (H) Wound area was analyzed at the indicated times after injury. The percentages of wound areas compared to original wound sizes are shown as a bar chart. Data are expressed as means ± SD. *P* = 0.0271 (day 5), 0.0072 (day 9) by unpaired two-tailed Student's *t* test. (*n* = 4 mice/group, biological replicates). (I) Representative images of wounds (5 days post-injury) of *Mrc1-DTR* mice treated with PDGF-AA or PBS for 5 days. Wound bed tissue samples were immunostained for EN1 (green) and Ki67 (red). White arrowheads indicate EN1$^+$Ki67$^+$ cells. Scale bars, 10 μm. Fluorescence images with an IgG-negative primary antibody control are available in the Fig. EV5. (J) The percentages of EN1$^-$, EN1$^+$Ki67$^-$, and EN1$^+$Ki67$^+$ cells out of the total DAPI-positive cells/0.01-mm$^2$ wound area were compared between mouse groups. Data are expressed as means ± SD. *P* = 0.0045 (EN1$^-$), 0.3201 (EN1$^+$ Ki67$^-$), 0.0076 (EN1$^+$ Ki67$^+$) by unpaired two-tailed *t* test after applying logit transformation. (*n* = 4 mice/group, biological replicates). Source data are available online for this figure.

125 μg/ml Liberase TL (Roche, Basel, Switzerland) in RPMI. Minced samples were incubated at 37 °C for 1 h 20 min with rotation, filtered through 70 μm and 40 μm filters, centrifuged, and resuspended in 2% FBS/PBS.

## Flow cytometry and cell sorting

Cells were stained with the following antibodies for 30 min on ice: CD45-Brilliant Violet 421 (Biolegend, CA, USA; 1:100), CD11b-PE/Cy7 (BD Pharmingen, CA, USA; 1:100), F4/80-PE (BD Pharmingen, CA, USA; 1:100), CD206-PE/Dazzle594 (Biolegend, CA, USA; 1:100), CD31-PE/Cy7 (BD Pharmingen, CA, USA; 1:100), CD26-FITC (Biolegend, CA, USA; 1:100). Immediately before sorting, 7AAD (BD Pharmingen, CA, USA) was added at a concentration of 1 μg/mL. Cell sorting was performed on a BD FACS Aria III with FACS DIVA software (BD Biosciences, CA, USA). FlowJo Software v10 (FlowJo, Ashland, OR) was used to analyze the flow cytometry data. In all conditions, the experiment was repeated three times. Statistical Analysis was performed using ANOVA followed by Tukey's post hoc test.

## Single-cell library generation

Live cells (7AAD-negative) were sorted from mouse wound tissue (3 mice/group) and processed for droplet-based scRNA-seq. Two technical replicates were performed, no biological replicates. Single-cell capture and library generation were performed using the Chromium Single Cell 3' Reagent Kit v3 (10x Genomics, CA, USA), according to the manufacturer's protocol. Sorted cells were resuspended at a concentration of approximately 1,000 cells/μL in PBS containing 0.04% BSA. We targeted 10,000 cells per sample for capture. For sample preparation, each single-cell suspension was mixed with RT-PCR master mix and loaded with Single Cell 3' v3 Gel Beads and Partitioning Oil onto Chromium Chip B. The chip was then loaded onto a Chromium Controller for single-cell GEM generation. Immediately following GEM generation, the Gel Beads were dissolved, primers were released, and any co-partitioned cells

were lysed. Incubation of the GEMs produced barcoded, full-length cDNA from poly-adenylated mRNA. After GEM disruption, full-length cDNA was amplified to generate a sufficient mass for library construction. Finally, amplified cDNAs were fragmented, and adapter and sample indexes were added to finished libraries. The size profiles of the pre-amplified cDNA and sequencing libraries were examined with an Agilent Bioanalyzer 2100 using a High Sensitivity DNA chip (Agilent Technologies, CA, USA). Libraries were sequenced on an Illumina Hiseq 2500 (Illumina, CA, USA).

## Analysis of scRNA-seq data

Sequencing data were analyzed using the Cell Ranger pipeline (v 6.0.0). Cell Ranger mkfastq was used to convert the barcode and read data to FASTQ files. Cell Ranger count was used to identify cell barcodes aligned to an indexed mm10 mouse genome. The count matrix data were analyzed using Seurat v4 and v5 (Hao et al, 2021) and in part using SCALA (Tzaferis et al, 2023). The data from all samples were combined, and an aggregate Seurat object was generated. To remove poor quality cells and doublet cells, we filtered cells that had more than 2000 unique features or 500 fewer, >6% mitochondrial RNAs. Smaller clusters expressing markers for multiple cell types were removed as doublets. After principal component analysis (PCA) was performed and a resolution of 0.2 was set, we obtained 12 clusters for the sample. These clusters were also identified based on the presence/absence of known marker genes of major cell types. To subcluster the macrophage population, the FindSubCluster function was used.

Marker genes of each cluster were identified by Wilcoxon rank-sum tests using the FindMarkers function of Seurat. For over-representation analysis, the top 200 upregulated genes with FDR <0.01 in each subpopulation were analyzed for enrichment with MSigDB hallmark and Reactome using decoupleR (Castanza et al, 2023; Subramanian et al, 2005).

Ligand–receptor interactions were analyzed using LIANA+ python package (Dimitrov et al, 2024; Dimitrov et al, 2022), using secreted signaling pairs of CellChat (Jin et al, 2021). *Plac8*$^{hi}$ and

*Crabp1*hi fibroblasts were combined as a *Gpnmb*lo fibroblast population. The interactions were analyzed using LIANA+'s rank aggregate function with default parameters, except for expr_prop, which was set to 0.05.

## RNA-seq

Poly-A mRNA was extracted from total RNA using a NEBNext poly(A) mRNA magnetic isolation module (New England Biolab), and RNA-seq libraries were prepared using a NEBNext Ultra RNA Library Prep kit for Illumina according to the manufacturer's protocol (New England Biolab). The libraries were then PCR-amplified for approximately 12 cycles and sequenced on a Novaseq (Illumina). Reads were aligned to the mm10 mouse genome using STAR (Dobin et al, 2013). Expression analysis of the RNA-seq data was performed using HOMER (Heinz et al, 2010). GSEA (Subramanian et al, 2005) was performed using rank files generated from expression data analyzed using DESeq2.

## Gene sets enriched in macrophage subpopulations

Over-representation analysis of the top 100 upregulated genes in each subpopulation, compared to the rest, with MSigDB hallmark (H) and Reactome (R) gene sets using decoupleR (Badia et al, 2022; Castanza et al, 2023; Subramanian et al, 2005). The top 10 terms with FDR <0.15 are shown. Top 200 gene lists of macrophage subclusters are also shown in Dataset EV2.

## Histology and immunofluorescence staining

Mouse back skin and wound beds were harvested for histology. Harvested tissues were fixed using Tissue-Tek Ufix (Sakura Finetek, Tokyo, Japan), embedded in paraffin, and cut into 10 μm-thick sections. Sections were then deparaffinized, rehydrated, and quickly washed in a 70, 80, 90, 95, and 100% ethanol series before finally washing twice in xylene. Images of hematoxylin and eosin (HE) and Masson-trichrome-stained sections were acquired using a BZ-X810 microscope (Keyence, Osaka, Japan) and analyzed to assess the thickness of granulation tissues. Granulation tissue thickness was measured at five points on each slide from three mice per group using ImageJ software.

For immunofluorescence staining, wound tissues were sub-merged in Tissue Tek OCT compound (Sakura Finetek, Tokyo, Japan) for 5 min at room temperature. Tissues were then embedded in Tissue Tek OCT under dry ice to achieve rapid freezing. Frozen blocks were mounted on a cryostat, and 6 μm-thick sections were transferred to slides. Fresh frozen sections (6 μm) were fixed in 4% paraformaldehyde for 5 min at room temperature. Next, slides were blocked for 1 h with 10% donkey serum prior to incubation with the following primary antibodies at 4 °C overnight: anti-CD206 (MCA2235GA, Bio-Rad, CA, USA; 1:200), anti-engrailed-1 (PA5-14149, Thermo Fisher Scientific, CA, USA; 1:200), anti-CD26 (AF954, R&D Systems, MN, USA; 1:200), anti-FAP (MAB9727, R&D Systems, MN, USA; 1:200) and anti-PDGF-A (SC-9974, Santa Cruz Biotechnology, CA, USA; 1:200). After washing, Slides were stained with the following secondary antibodies at room temperature for 1 h: Alexa Fluor 488 conjugated anti-rabbit (A-11070, Invitrogen, CA, USA; 1:500), Alexa Fluor 546 conjugated anti-rat (A-11081, Invitrogen, CA,

USA; 1:500), and Alexa Fluor 647 conjugated anti-mouse (115-607-187, Jackson ImmunoResearch, PA, USA; 1:500) antibodies. Finally, slides were stained with DAPI. Immunofluorescence images were obtained using a BZ-X810 microscope (Keyence) at 4, 10, 20 and 40 x magnification. Quantitative analysis of labeled cells was based on the methods described previously (Junankar et al, 2006). Briefly, cells were counted in three random high-power fields per wound area in tissue sections from three mice per group using ImageJ software.

## Quantitative RT-PCR

Total RNA was isolated using Nucleospin RNA (Macherey-Nagel, Düren, Germany), according to the manufacturer's instructions. Complementary DNA (cDNA) was synthesized using ReverTra Ace qPCR RT Master Mix with genomic DNA (gDNA) Remover (TOYOBO). All qPCR analyses were performed with QuantStudio 5 using the Taqman assay system (Thermo Fisher Scientific, CA, USA). Values obtained by the ddCt method were normalized to the expression of *Gapdh* and then further normalized to the values in the control samples.

## Primary mouse fibroblast culture

Culture of mouse newborn skin fibroblasts was done as previously described (Reiisi et al, 2010). In brief, newborn mice were sacrificed by decapitation and their trunk skin was removed with forceps. After removing the fat, the skin was sliced into small pieces. Skin pieces were then placed into a 60 mm cell culture dish containing 4 ml fibroblast growth medium (DMEM supplemented with 10% FBS, 1% Penicillin/Streptomycin and 50 μg/mL L-Ascorbic Acid Phosphate). Cells were incubated at 37 °C in the presence of 5% $CO_2$ and treated with PDGF-AA as previously described (Juhl et al, 2020). For experiments, primary mouse fibroblasts were seeded on gelatin-coated 48-well plates at 30,000 cells per well in fibroblast growth medium. After 24 h, the culture medium was replaced with medium containing 1% FBS and incubated for an additional 24 h. The cells were then treated with 40 ng/mL mouse PDGF-AA (BioLegend, catalog number 776302) for 7 days. PDGF-AA was prepared by dissolving it in PBS containing 0.2% BSA. During the 7-day culture period, the medium was refreshed on days 3 and 5.

## Treatment of mice with PDGF-AA

To evaluate the effects of topically administered PDGF-AA on wound healing, *Mrc1-DTR* mice with standard splinted wounds were used. Treatment with PBS and recombinant PDGF-AA was performed as described previously (Demaria et al, 2014). Mice were treated with PDGF-AA (20 ng topical application) or PBS (control) for 10 days, starting 1 day after wounding. Each wound site was photographed on a fixed day, and the relative wound area was calculated as wound area divided by the area of the splint hole.

## Human keloid samples

Surgically excised keloid tissues from the earlobes of three patients were utilized for this study. All samples were subjected to

immunohistochemical analysis. The cohort consisted of two males and one female, all of Asian descent, with a mean age of 24.7 years.

## Immunostaining of human keloid samples

The surgically excised human earlobe keloid tissues were fixed in 10% formalin, embedded in paraffin, and sliced into 8 μm cross sections. Specimens were subjected to HE staining and immunostaining. The antibodies used were as follows: anti-mannose receptor antibody (Abcam, ab64693) and anti-engrailed-1 antibody (Bioss, BS-11744R).

## Statistical analysis

Data are presented as mean ± SD except where otherwise indicated. Sample sizes were not based on power calculations. Statistical significance was determined using the two-tailed Student's $t$ test. Two-way ANOVA followed by post hoc Tukey's post hoc test was used for experiments involving two factors, except where otherwise indicated. Significant values are indicated as $*P < 0.05$. All statistical analyses were performed using Prism software (GraphPad, San Diego, CA, USA).

## Ethics declarations

This study was conducted in accordance with the principles of the Declaration of Helsinki and approved by the Institutional Ethics Committee of Nippon Medical School Hospital, Tokyo, Japan (B-2024-894). All study participants provided written informed consent.

# Data availability

The datasets presented in this study can be found online in the GEO database under accession number GSE268684.

The source data of this paper are collected in the following database record: biostudies:S-SCDT-10_1038-S44319-025-00496-4.

# Peer review information

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

## Acknowledgements

The authors thank Noriko Yamanaka and Eriko Magoshi for their excellent technical assistance. This study was supported by the Japan Society for the Promotion of Science (JSPS) KAKENHI grant numbers JP22K19534, JP 23H02912, JP 23H02901 and JP 24K22112; Japan Science and Technology Agency (JST) grant number JPMJMS2023 and JPMJFR2351; Japan Agency for Medical Research and Development (AMED) under grant numbers JP20gm6210023, and JP21bm0704045; and by the Takeda Science Foundation.

## Author contributions

**Azusa Honda**: Conceptualization; Data curation; Investigation; Writing—original draft. **Hiroyuki Koike**: Data curation; Formal analysis; Investigation; Methodology; Writing—review and editing. **Teruyuki Dohi**: Data curation; Formal analysis. **Eri Toyohara**: Investigation. **Sumio Hayakawa**: Supervision. **Kazuyuki Tobe**: Resources. **Ichiro Manabe**: Visualization; Writing—review and editing. **Rei Ogawa**: Supervision. **Yumiko Oishi**: Conceptualization; Formal analysis; Supervision; Writing—original draft; Writing—review and editing.

Source data underlying figure panels in this paper may have individual authorship assigned. Where available, figure panel/source data authorship is listed in the following database record: biostudies:S-SCDT-10_1038-S44319-025-00496-4.

## Disclosure and competing interests statement

The authors declare no competing interests.

# Expanded View Figures

**Figure EV1.  Flow cytometric analysis of wound tissue.**

Wound tissues were harvested from C57BL/6 mice at the indicated times after injury and analyzed by flow cytometry. (**A**) Gating strategy. Live cells were sub-gated based on CD45 and CD11b expression, and CD45$^+$CD11b$^+$ myeloid cells were sub-gated based on the expression of Ly6G. Ly6G$^-$ cells were further sub-gated based on CD206 and F4/80 expression to reveal the CD206$^+$F4/80$^+$ population. CD45$^-$CD11b$^-$ cells were sub-gated based on CD31 and Epcam expression. CD31$^-$Epcam$^-$CD26$^+$ cells were defined as fibroblasts. The diagram for the control mice in Fig. 1B is presented again as a CD206 and F4/80 gating plot to show the correspondence. (**B**) Isotype control of the wound macrophages. The isotype control for wound macrophages was used to assess CD206 and F4/80 expression. The cells were stained with the respective antibodies, and instrument settings and compensation were adjusted using the corresponding isotype control. The analysis was performed using FlowJo software ($n = 3$ wounds, biological replicates). The diagram for the control mice in Fig. 1B is presented again to show the correspondence. (**C**) Fluorescent minus one (FMO) control for the CD206 and F4/80 staining to identify and gate cells in flow cytometry experiments ($n = 3$ wounds, biological replicates). (**D–F**) Temporal changes in the numbers of fibroblasts and macrophages in wounds. The number of CD206$^+$ macrophages per wound over time is shown (**D**). Data are shown as means ± SD. *$p = 0.0008$ (day 3), 0.0002 (day 5), 0.0685 (day 7), 0.9886 (day 14) by one-way ANOVA followed by Dunnett's multiple comparisons test ($n = 3$ mice/group, biological replicates). The number of CD26$^+$ fibroblasts at 5 days post injury (**E**) and the proportion of CD45$^+$ cells relative to the total number of live cells (**F**) in our mouse skin injury model are shown. Data are shown as means ± SD. *$P = 0.0285$ (**E**), 0.0493 (**F**) by unpaired two-tailed Student's $t$ test ($n = 3$ mice/group, biological replicates).

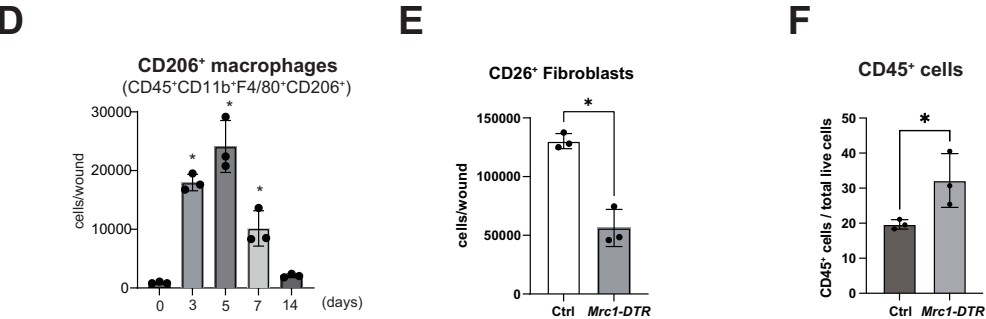

**A**

**B**

Ctrl    *Mrc1-DTR*    Isotype control

F4/80    CD206

**C**

All stained

FMO (F4/80)

FMO (CD206)

F4/80    CD206

**D**

CD206⁺ macrophages
(CD45⁺CD11b⁺F4/80⁺CD206⁺)

cells/wound

0  3  5  7  14  (days)

**E**

CD26⁺ Fibroblasts

cells/wound

Ctrl    *Mrc1-DTR*

**F**

CD45⁺ cells

CD45⁺ cells / total live cells

Ctrl    *Mrc1-DTR*

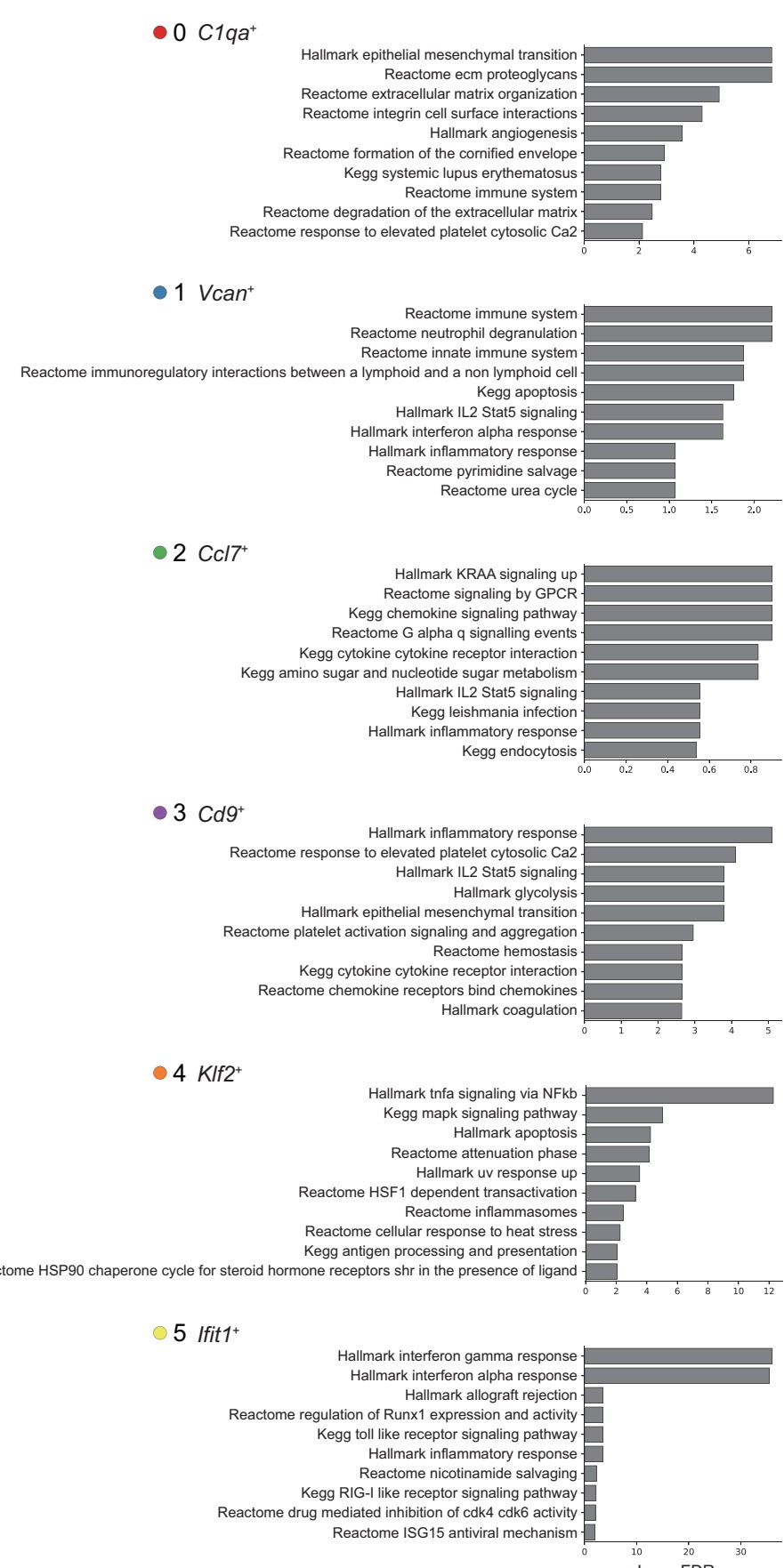

**Figure EV2.  Gene sets enriched in highly expressed genes in macrophage subpopulations.**

Over-representation analysis of the top 100 upregulated genes in each subpopulation, compared to the rest, with MSigDB hallmark, Reactome, and Kegg gene sets using decoupleR (Badia et al, 2022; Castanza et al, 2023; Subramanian et al, 2005). The top 10 terms are shown. A list of genes used in the analysis is presented in Dataset EV2.

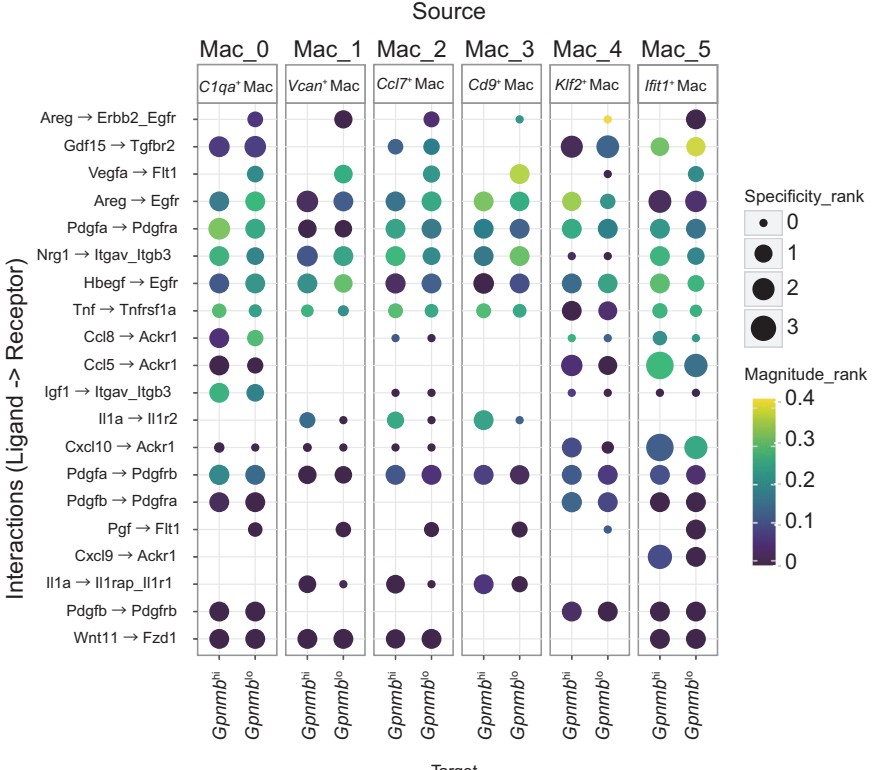

**Figure EV3. Dot plot of ligand–receptor-interactions between macrophage subpopulations and *Gpnmb*^hi/*Gpnmb*^lo fibroblasts.**

Top 20 ligand–receptor pairs of the aggregate consensus rank are shown. Macrophage subpopulations expressing ligands and fibroblast subpopulations expressing receptors are shown on the x axis. Ligand and cognate receptor combinations are shown on the y axis. Circle size denotes specificity rank, a measure of how specific an interaction is to a given pair of cell groups; color denotes magnitude rank showing the strength of ligand–receptor interaction.

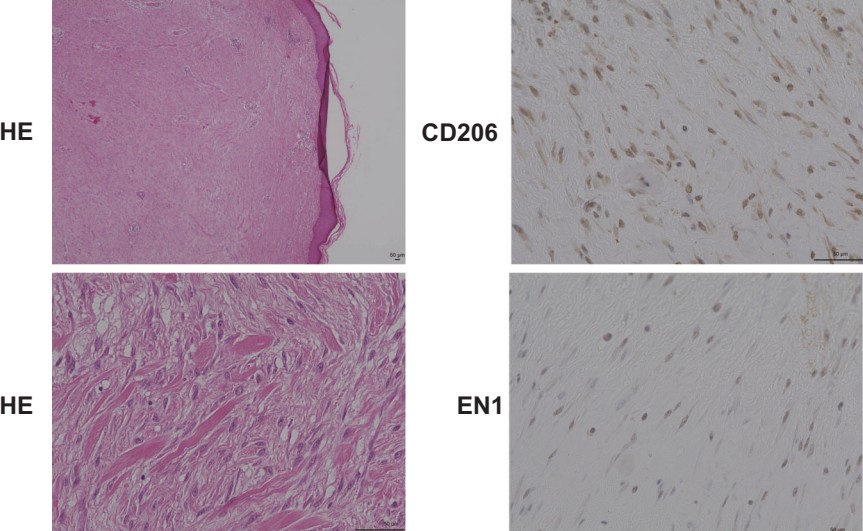

**Figure EV4. Presence of CD206⁺ macrophages and EN1⁺ fibroblasts in human keloid tissue.**

Representative histologic images of human keloid tissue stained with hematoxylin and eosin (HE) and immunostained for CD206 and EN1. Scale bars: 50 μm.

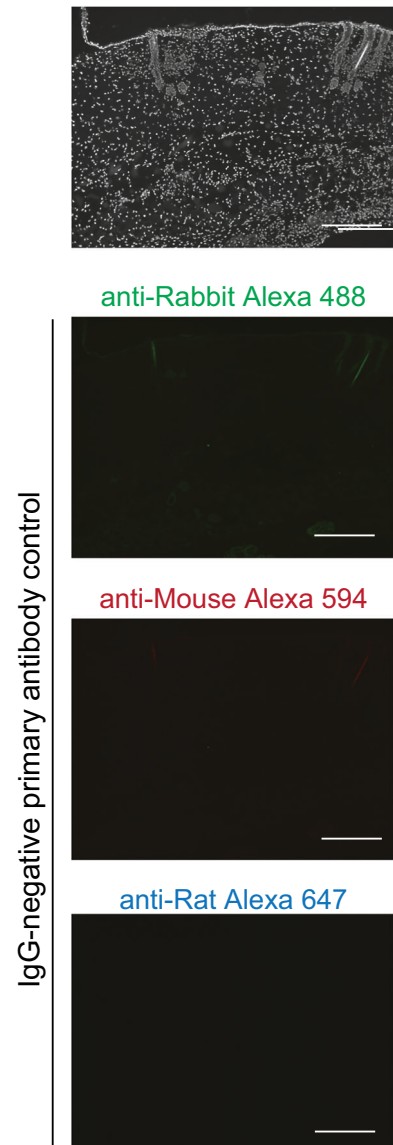

**Figure EV5.   Negative control for immunofluorescence staining for Fig. 5C,G.**

Tissue sections were stained with secondary antibodies only to confirm the absence of non-specific fluorescence signals. Images show DAPI (white), Alexa Fluor 488 anti-Rabbit (green), Alexa Fluor 594 anti-Mouse (red), and Alexa Fluor 647 anti-Rat (blue). Scale bar: 200 μm.

                                                                 