## [Peer Review File · EMBO Reports]

CD206+ macrophages facilitate wound healing through interactions with *Gpnmb*^{hi} fibroblasts

Azusa Honda, Hiroyuki Koike, Teruyuki Dohi, Eri Toyohara, Sumio Hayakawa, Kazuyuki Tobe, Ichiro Manabe, Rei Ogawa, and Yumiko Oishi

Corresponding author(s): Yumiko Oishi (oishi.yumiko@tmd.ac.jp)

Review Timeline:

Submission Date:	22nd Oct 24
Editorial Decision:	28th Nov 24
Revision Received:	10th Mar 25
Editorial Decision:	3rd Apr 25
Revision Received:	22nd Apr 25
Accepted:	27th May 25

Editor: Achim Breiling

Transaction Report:

Dear Dr. Oishi,

Thank you for the submission of your manuscript to EMBO reports. I have now received the reports from the three referees that were asked to evaluate your study, which can be found at the end of this email.

As you will see, the referees think that these findings are of interest. However, they have several comments, concerns, and suggestions, indicating that a major revision of the manuscript is necessary to allow publication of the study in EMBO reports. As the reports are below, and all the referee concerns need to be addressed, I will not detail them here.

Given the constructive referee comments, I would like to invite you to revise your manuscript with the understanding that the concerns of the referees must be addressed in the revised manuscript and in a detailed point-by-point response. Acceptance of your manuscript will depend on a positive outcome of a second round of review. It is EMBO reports policy to allow a single round of revision only and acceptance of the manuscript will therefore depend on the completeness of your responses included in the next, final version of the manuscript.

- 1) a .docx formatted version of the final manuscript text (including legends for main figures, EV figures and tables), but without the figures included. Figure legends should be compiled at the end of the manuscript text.
- 2) individual production quality figure files as .eps, .tif, .jpg (one file per figure), of main figures and EV figures. Please upload these as separate, individual files upon re-submission.

- 4) a complete author checklist, which you can download from our author guidelines (<https://www.embopress.org/page/journal/14693178/authorguide>). Please insert page numbers in the checklist to indicate where the requested information can be found in the manuscript. The completed author checklist will also be part of the RPF.

- 5) that primary datasets produced in this study (e.g. RNA-seq, ChIP-seq, structural and array data) are deposited in an

appropriate public database. If no primary datasets have been deposited, please also state this in a dedicated section (e.g. 'No primary datasets have been generated and deposited'), see below.

The accession numbers and database should be listed in a formal "Data Availability" section that follows the model below. This is now mandatory (like the COI statement). Please note that the Data Availability Section is restricted to new primary data that are part of this study. This section is mandatory. As indicated above, if no primary datasets have been deposited, please state this in this section

Data availability

8) Regarding data quantification and statistics, please make sure that the number "n" for how many independent experiments were performed, their nature (biological versus technical replicates), the bars and error bars (e.g. SEM, SD) and the test used to calculate p-values is indicated in the respective figure legends (also for EV and Appendix figures). Please also check that all the p-values are explained in the legend, and that these fit to those shown in the figure. Please provide statistical testing where applicable. Please avoid the phrase 'independent experiment', but clearly state if these were biological or technical replicates. Please also indicate (e.g. with n.s.) if testing was performed, but the differences are not significant. In case n=2, please show the data as separate datapoints without error bars and statistics. See also: <http://www.embopress.org/page/journal/14693178/authorguide#statisticalanalysis>

9) Please add scale bars of similar style and thickness to microscopic images, using clearly visible black or white bars (depending on the background). Please place these in the lower right corner of the images themselves. Please do not write on or near the bars in the image but define the size in the respective figure legend.

10) Please also note our reference format:

12) We now use CRediT to specify the contributions of each author in the journal submission system. CRediT replaces the author contribution section. Please use the free text box to provide more detailed descriptions and do NOT provide your final manuscript text file with an author contributions section. See also our guide to authors: <https://www.embopress.org/page/journal/14693178/authorguide#authorshipguidelines>

13) All Materials and Methods need to be described in the main text using our 'Structured Methods' format, which is required for

all research articles. According to this format, the Methods section should include a Reagents and Tools Table (listing key reagents, experimental models, software, and relevant equipment and including their sources and relevant identifiers), uploaded as separate file, and a Methods section in which we encourage the authors to describe their methods using a step-by-step protocol format with bullet points, to facilitate the adoption of the methodologies across labs. More information on how to adhere to this format as well as downloadable templates (.doc) for the Reagents and Tools Table can be found in our author guidelines (section 'Structured Methods'):

14) Please order the manuscript sections like this, using these names:

Title page - Abstract - Keywords - Introduction - Results - Discussion - Methods - Data availability section - Acknowledgements (including funding information) - Disclosure and Competing Interests Statement - References - Figure legends - Expanded View Figure legends

15) Please make sure that all the funding information is also entered into the online submission system and that it is complete and similar to the one in the acknowledgement section of the manuscript text file.

I look forward to seeing a revised form of your manuscript when it is ready.

Yours sincerely,

Referee #1:

The paper covers and extremely interesting topic, but perhaps the mechanisms underlying the communication between (myo)fibroblasts and macrophages could be better developed.

In particular, a recent paper by Ezzo et al published in Sci Adv (<https://pubmed.ncbi.nlm.nih.gov/39441936/>) could be used as a basis for expansion of the study. For example, is there an activation of YAP and induction of the YAP targets CCN1 and CCN2 that is dependent on CD206+ macrophages?

Referee #2:

Overall Rating:

The manuscript contains original new data that provides a more detailed analysis of how CD206+ cells impact wound healing by applying scRNAseq data analysis. In addition, the linkage between these cells and stimulation of PDGF α was experimentally verified by treatment and subsequent improved healing. The data generally supports the hypothesis and the manuscript overall is well written and of appropriate length.

Suitability for publication:

While the data warrants further consideration the manuscript has deficiencies in its present form. A notable limitation is the modest impact of CD206+ cell depletion, potentially attributable to the relatively high numbers of CD206+ cells that remained after DT treatment. This limitation should be explicitly addressed in both the Abstract and Discussion. Additionally, there are several experimental and analytical aspects that could be improved, as outlined below.

Major Comments:

1. The difference in healing associated with Mrc1-deleted cells, while statistically significant, is clinically modest. This limitation should be discussed in the Abstract and Discussion.
2. Figure 1B is missing fluorescence-minus-one (FMO) controls for F4/80 and CD206, which are needed to confirm specificity. The double-positive cell pattern, aligned at a 45-degree angle, suggests non-specific binding. These controls should be included.
3. The >2-fold increase in immune cells observed in the single-cell RNA sequencing data should be validated using flow cytometry.
4. More details on quality control parameters and thresholds used to exclude apoptotic cells, doublets, and ghost cells from the dataset should be provided.
5. Figure 5 immunofluorescence data has issues with specificity and clarity. An IgG-negative primary antibody control should be included to validate specificity. Background noise obscures specific staining in PDGFR (Fig. 5C) and Ki67 (Fig. 5G). Quantitative data is missing for Figs. 5B and 5C, limiting analysis. The sample size and exact post-wounding time points for the images should also be specified.
6. The sub-cluster analysis of macrophages in Extended Data Figure 3 is incomplete. The overall phenotype of macrophage clusters should be tied more directly to wound healing. A table summarizing the percentages of macrophage clusters by condition in the main figures (not just in Extended Data) would be helpful. Differences in sub-clustering by condition should also explain the reduction in F4/80+CD206+ cells in Mrc1-DTR mice (Fig. 1C) alongside the increase in F4/80+ macrophages in the same model (Figs. 1H and 1I).

Minor Comments:

1. The Abstract should report the magnitude of changes and statistical significance.
2. Specify the source of PDGF-AA (company name, catalog number) and whether it is ready-to-use or requires dilution. If dilution is necessary, include details on the solvent, final concentration, and treatment frequency.
3. Figure 1A lacks images showing the gross appearance of the mice.
4. Clarify whether Table 1C applies to all cells or only CD206+F4/80+ cells. The Y-axis should reflect this distinction.
5. Since Gpnmbhi fibroblasts are most affected by CD206+ depletion, provide more information such as a table with differential gene expression (DGE) for these fibroblasts compared to others. It would also be useful to analyze DGE by condition to determine how their phenotype is influenced by CD206+ depletion. Figure 3E provides limited insight into their state.
6. The differential gene expression results from FindMarkers in Fig. 2C should be included as a supplemental table.
7. Discuss the limitations of DT-induced CD206+ cell depletion, particularly the potential effects on other cell types expressing CD206 and how this might influence wound healing outcomes.
8. Immunofluorescence and histological analyses lack sufficient experimental details, such as the concentration of blocking media, primary and secondary antibodies, fixation duration, image magnification, and the number of replicates.
9. The flow cytometry and cell sorting sections are missing details about blocking buffers, antibody concentrations, and the number of replicates, as well as the statistical tests applied.
10. There are several grammatical errors throughout the manuscript. Careful proofreading is required.

Referee #3:

****Title:**** The title could be refined to better align with the research question. The study investigates the interaction between macrophages and fibroblasts with the aim of preventing fibrosis and scarring. A revised title could be: "Crosstalk between Macrophages CD260 and Gpnmb Fibroblasts is Essential for Effective Skin Regeneration."

****Introduction:**** The research question is clearly articulated; however, the evidence analysis could be strengthened to provide more robust support for the central research inquiry.

--In the introduction, it would be necessary to mention a short paragraph about F4/80+CD206+CD301+ (see Shook et al., 2016). It would also be nice to summarize the effect of the depletion of macrophages at the early and intermedia stages of previous reports.

****Methodology:**** The chosen research methodology is suitable for addressing the research question. While the data collection methods are reliable and valid, a more detailed explanation of the data and experimental design would enhance clarity and comprehension.

- Why do you inject DT every 2 days? Do the myeloid cells recover within 2 days?

-Fig 1. Why was the RNAseq performed on day 7? All previous experiments were performed on day 5. The authors didn't analyze the effect during the early state and cytokines expression. There is no analysis of pro-inflammatory macrophages, specifically F4/80+CD80+. The results do not support their conclusion about significantly attenuating re-epithelization, so I strongly recommend additional experiments. It would be nice to compare the wound closure between WT and the Mrc1-DT group. I am having difficulty comprehending which specific sections of the wound are represented in the images provided at high magnifications.

-Fig 4. To provide more robust support for the conclusion, I recommend conducting additional experiments, including co-staining with markers for different layers. It would be nice to add the representative Trichrome image of the wound. I am having difficulty comprehending which specific sections of the wound are represented in the pictures provided at high magnifications.

-Fig 5. I struggle to discern the notable differences between groups; the PDGF-A signal appears unspecific in the

immunostaining. To provide more robust support for the effect of the PDGF-AA treatment, I recommend conducting additional experiments. Again, I struggle to discern the notable differences between groups (Fig 5G.)

****Results and Discussion****: The findings must be interpreted and discussed more in-depth concerning the research question and previous antecedents. Additionally, the authors could further enhance the paper's impact by discussing the implications of their findings beyond Mrc1-DTR transgenic mice and exploring potential avenues for future research in this area. The paper doesn't acknowledge the study's limitations and possible areas for further research.

I recommend that the authors conduct a comprehensive revision of their manuscript to enhance its clarity and depth before considering resubmission for publication.

Response to Referees

We thank the Referees for their thoughtful comments on our manuscript. We greatly appreciate the effort and time taken to provide us with a number of constructive comments, which have helped us significantly improve the quality of our article. (Excerpts from the referee's critique are indicated in bold.)

Referee #1:

The paper covers an extremely interesting topic, but perhaps the mechanisms underlying the communication between (myo)fibroblasts and macrophages could be better developed. In particular, a recent paper by Ezzo et al published in Sci Adv (<https://pubmed.ncbi.nlm.nih.gov/39441936/>) could be used as a basis for expansion of the study. For example, is there an activation of YAP and induction of the YAP targets CCN1 and CCN2 that is dependent on CD206+ macrophages?

We thank the reviewer for the insightful suggestion. As suggested, we analyzed expression levels of YAP/TAZ target genes, *Ccn1*, *Ccn2*, *Ankrd1*, and (PubMed ID 39441936 and 35191398) in the scRNA-seq dataset. We found that expression of *Ccn1* and *Ccn2* was not detected and *Ankrd1* UMI counts were too low for meaningful comparison, probably due to the technical limitations inherent to the 10X platform. Regarding *Serpine1*, diphtheria toxin-mediated depletion of CD206⁺ cells actually increased its expression levels as shown in the accompanying graph.

We then further analyzed expression of *Ccn1* and *Ccn2* in the bulk RNA-seq dataset of the wounded tissues on day 5 (Fig. 1J). Although *Ccn1* expression level tended to be decreased in *Mrc1*-DTR wounds, the difference did not reach statistical significance ($P = 0.34$). *Ccn2* levels were not changed. Because the wounds contained varying cellular components, these data may not strictly reflect the changes of these genes in fibroblasts.

Taken together, the analysis of the YAP/TAZ target genes in our scRNA-seq and bulk RNA-seq datasets is inconclusive. To clearly define the role of CD206⁺ macrophages in fibroblast YAP/TAZ signaling, future studies will need to employ multiple modalities to assess signaling activity (e.g., direct analysis of YAP/TAZ protein levels or localization). We have added discussion on YAP/TAZ signaling as a potential mechanism mediated by CD206⁺ macrophages (p. 23). We would like to note that because the silicon ring-placement model inhibits contraction of myofibroblasts, it may interfere with the YAP/TAZ signaling (p. 28). That said, He et al. (PubMed ID 35191398) recently showed that fibroblast-specific YAP/TAZ deficiency inhibited fibrosis in the kidney, liver, and lung; it is likely that fibroblast YAP/TAZ signaling is also important in skin wound healing. We will need to address how YAP/TAZ signaling is regulated in skin fibroblasts after skin injury. Thank you again for your valuable comments.

Response to Referee #2:

Overall Rating:

The manuscript contains original new data that provides a more detailed analysis of how CD206⁺ cells impact wound healing by applying scRNAseq data analysis. In addition, the linkage between these cells and stimulation of PDGF α was experimentally verified by treatment and subsequent improved healing. The data generally supports the hypothesis and the manuscript overall is well written and of appropriate length.

Suitability for publication:

While the data warrants further consideration the manuscript has deficiencies in its present form. A notable limitation is the modest impact of CD206⁺ cell depletion, potentially attributable to the relatively high numbers of CD206⁺ cells that remained after DT treatment. This limitation should be explicitly addressed in both the Abstract and Discussion. Additionally, there are several experimental and analytical aspects that could be improved, as outlined below.

Major Comments:

1. The difference in healing associated with *Mrc1*-deleted cells, while statistically significant, is clinically modest. This limitation should be discussed in the Abstract and Discussion.

We have addressed this by revising Abstract and Discussion as follows:

Abstract: “selective depletion of CD206⁺ macrophages results in modest but significant delays in wound healing (p.2).”

Discussion: “although the differences were statistically significant, the magnitude of inhibition of healing processes in *Mrc1-DTR* mice was modest. One potential reason for the modest effect is that the administration of diphtheria toxin only partially depleted CD206⁺ cells. It is also possible that the remaining CD206⁺ macrophages may alter their functions particularly at later time points, since wound tissue microenvironment changes over the course of inflammation to

healing. As such, we need to further analyze spatiotemporal changes in cell subpopulations and their dynamic interactions throughout the repair process (p.28).”

2. Figure 1B is missing fluorescence-minus-one (FMO) controls for F4/80 and CD206, which are needed to confirm specificity. The double-positive cell pattern, aligned at a 45-degree angle, suggests non-specific binding. These controls should be included.

Thank you for pointing out the need for fluorescence-minus-one (FMO) controls to confirm the specificity of F4/80 and CD206 staining. We have included FMO controls for F4/80 and CD206 in our analysis (EV Figure 1C), which confirm the specificity of the F4/80 and CD206 staining and allow us more precise gating of positivity.

3. The >2-fold increase in immune cells observed in the single-cell RNA sequencing data should be validated using flow cytometry.

Thank you for your suggestion to validate the increase in immune cells in wounds. We have now performed flow cytometry analysis and measured the proportion of CD45⁺ cells within the total live cells. The results also show approximately 1.7-fold increase in CD45⁺ immune cells (EV Figure 2C), supporting the observed increase in the immune cell populations in the scRNA-seq dataset.

4. More details on quality control parameters and thresholds used to exclude apoptotic cells, doublets, and ghost cells from the dataset should be provided.

We have added detailed descriptions of the quality control metrics used, including parameters for mitochondrial gene content, UMI counts, and doublet identification methods in the "Materials and Methods" section (p.35).

5. Figure 5 immunofluorescence data has issues with specificity and clarity. An IgG-negative primary antibody control should be included to validate specificity. Background noise obscures specific staining in PDGFR (Fig. 5C) and Ki67 (Fig. 5G). Quantitative data

is missing for Figs. 5B and 5C, limiting analysis. The sample size and exact post-wounding time points for the images should also be specified.

As suggested, we added images of the IgG primary antibody control for the PDGFR and Ki67 staining to Appendix Figure S2. We quantitated the results of Figures 6B and 6D, which correspond to Figures 5B and 5C in the previous versions of the manuscript, and showed in Figures 6C and 6E. We also described the sample size and post-wounding time points in the figure legend (p. 49). We thank the reviewer for these suggestions that strengthen the presentation of our findings.

6. The sub-cluster analysis of macrophages in Extended Data Figure 3 is incomplete. The overall phenotype of macrophage clusters should be tied more directly to wound healing. A table summarizing the percentages of macrophage clusters by condition in the main figures (not just in Extended Data) would be helpful. Differences in sub-clustering by condition should also explain the reduction in F4/80+CD206+ cells in Mrc1-DTR mice (Fig. 1C) alongside the increase in F4/80+ macrophages in the same model (Figs. 1H and 1I).

As suggested, we moved the original Supplementary Fig. 3 to main Fig. 3. We have also expanded characterization of subpopulations of macrophages and now show marker gene expression in subpopulations (Fig. 3D) and the plots showing results of overrepresentation analysis (EV Fig. 3). As shown in Fig 3D, *Adgre1* (F4/80) and *Mrc1* (CD206) expression was higher in clusters 0, 2, 4, and 5, and lower in clusters 1 and 3. In *Mrc1-DTR*, cluster 0, which showed the high level expression of *Adgre1* and *Mrc1*, was much reduced, while cluster 1 and 3, which showed lower level expression of *Adgre1* and *Mrc1* were increased. Accordingly, the changes in macrophage subpopulations are in agreement with the findings of Fig. 1C, E, H, and I, which show relative increase in F4/80⁺ cells, while CD206⁺ cells were decreased.

Minor Comments:

1. The Abstract should report the magnitude of changes and statistical significance.

We have revised the abstract to include the magnitude of changes and statistical significance.

2. Specify the source of PDGF-AA (company name, catalog number) and whether it is ready-to-use or requires dilution. If dilution is necessary, include details on the solvent, final concentration, and treatment frequency.

PDGF-AA was purchased from Biolegend (Catalog Number: 776308). It requires dilution in PBS to a final concentration of 40 ng/mL and was added to the culture media. This information was also included in the methods section in the revised manuscript (p. 36).

3. Figure 1A lacks images showing the gross appearance of the mice.

We have added images to Figure 1A showing the gross appearance of the mice.

4. Clarify whether Table 1C applies to all cells or only CD206⁺F4/80⁺ cells. The Y-axis should reflect this distinction.

Figure 1C applies specifically to CD45⁺CD11b⁺F4/80⁺CD206⁺ macrophages per wound. We have corrected the Y-axis label to reflect this distinction.

5. Since *Gpnmb*^{hi} fibroblasts are most affected by CD206⁺ depletion, provide more information such as a table with differential gene expression (DGE) for these fibroblasts compared to others. It would also be useful to analyze DGE by condition to determine how their phenotype is influenced by CD206⁺ depletion. Figure 3E provides limited insight into their state.

As suggested, we analyzed differential gene expression for *Gpnmb*^{hi}, *Plac8*^{hi}, and *Crabp1*^{hi} fibroblast populations, comparing conditions with and without DTR (Appendix Figure S1). Cholesterol homeostasis and mTORC1 signaling were upregulated in fibroblast populations in all three fibroblast populations after CD206⁺ macrophage depletion. The cell cycle-related gene sets such as G2M signaling, E2F targets, and p53 pathway were also differentially enriched in *Mrc1-DTR Plac8*⁺ fibroblasts, suggesting their potential compensatory proliferation. However,

we clearly need to further analyze whether their proliferation is activated by CD206⁺ macrophage depletion. Therefore, we do not state this possibility in the main text.

Our analysis revealed that while the effects of CD206⁺ macrophage depletion were evident in the altered composition of fibroblast subpopulations (particularly the decrease in *Gpnmb*^{hi} fibroblasts), the transcriptomic changes within each subpopulation were relatively modest. Because clustering, which we performed on all cells, already grouped the cells based on transcriptomic similarities, the difference caused by CD206⁺ may mainly exhibit as population changes. Accordingly, we only described that CD206⁺ macrophage depletion might have altered cellular metabolism across the fibroblast populations (p. 19).

6. The differential gene expression results from FindMarkers in Fig. 2C should be included as a supplemental table.

We have included the differential gene expression results from FindMarkers in Figure 2C, as an EV table 1 in the revised manuscript.

7. Discuss the limitations of DT-induced CD206⁺ cell depletion, particularly the potential effects on other cell types expressing CD206 and how this might influence wound healing outcomes.

We looked at the expression of CD206 in all cell types and added in Fig. 2D. The dominant cell population expressing *Mrc1* was macrophages (Fig. 2D), scRNA-seq data showed that dendritic cells and a part of neutrophils also expressed *Mrc1*, suggesting that diphtheria toxin induced cell death also in the latter cells. However, among immune cells, the fraction of neutrophils was increased, presumably reflecting prolonged inflammation. We have clearly stated the limitation of the DT-induced CD206⁺ cell depletion in Discussion particularly the potential effects on other cell types expressing CD206 might influence wound healing (p. 28).

8. Immunofluorescence and histological analyses lack sufficient experimental details, such as the concentration of blocking media, primary and secondary antibodies, fixation duration, image magnification, and the number of replicates.

We have added the experimental details in the Materials and Method section (p. 38).

9. The flow cytometry and cell sorting sections are missing details about blocking buffers, antibody concentrations, and the number of replicates, as well as the statistical tests applied.

We have added the experimental details in the Materials and Method section (p. 33).

10. There are several grammatical errors throughout the manuscript. Careful proofreading is required.

We have carefully proofread the manuscript to correct the grammatical errors. We thank the reviewer for rigorous review.

Response to Referee #3:

****Title:** The title could be refined to better align with the research question. The study investigates the interaction between macrophages and fibroblasts with the aim of preventing fibrosis and scarring. A revised title could be: "Crosstalk between Macrophages CD260 and Gpmb Fibroblasts is Essential for Effective Skin Regeneration."**

Thank you for your insightful suggestion. We have refined the title as "Crosstalk between CD206⁺ Macrophages and Gpmb^{hi} Fibroblasts is Essential for Effective Skin Regeneration" to better state research question and findings.

****Introduction:** The research question is clearly articulated; however, the evidence analysis could be strengthened to provide more robust support for the central research inquiry.**

--In the introduction, it would be necessary to mention a short paragraph about F4/80⁺CD206⁺CD301⁺ (see Shook et al., 2016). It would also be nice to summarize the effect of the depletion of macrophages at the early and intermedia stages of previous reports.

We have included a short paragraph about the function of F4/80⁺CD206⁺CD301⁺ macrophages, as reported previously (Shook et al., 2016) in the introduction (p. 6). As suggested by reviewer 3, we have included the effect of macrophage depletion in the Introduction. Previous reports have shown that macrophages play different roles at different stages of wound healing. Early depletion (days 0-1) reduces the number of fibroblasts without significantly affecting healing, while depletion from day 2 impairs re-epithelialization, vascularization, and fibroblast proliferation. Mid-stage depletion (days 3-7) severely disrupts healing. Ly6C^{hi} macrophages prepare tissues for repair, while Ly6C^{lo} macrophages, which transition on days 2-3, are critical for regeneration, highlighting their complementary roles in skin injury recovery (p.5).

****Methodology:** The chosen research methodology is suitable for addressing the research question. While the data collection methods are reliable and valid, a more detailed**

explanation of the data and experimental design would enhance clarity and comprehension.

- Why do you inject DT every 2 days? Do the myeloid cells recover within 2 days?

Thank you for your valuable comment. Though tissue macrophages may have a long life-span [van Furth R, Cohn ZA. The origin and kinetics of mononuclear phagocytes. *J Exp Med*. 1968;128:415–435. doi: 10.1084/jem.128.3.415.], particularly during the early-stage monocytes are recruited to the wound and differentiate into macrophages, including CD206⁺ macrophages. CD206⁺ macrophages also proliferate and increase in the number after injury. Accordingly, we injected DT every other day to ensure continuous reduction of CD206⁺ cells. Previous studies have shown that CD206⁺ cells were successfully depleted without recovery in the injured muscle by injecting DT every 2 days (Nawaz et al. *Nat Commun* 13: 7058). Therefore, the same method was chosen to maintain the depletion of CD206⁺ cells throughout the experimental period.

-Fig 1. Why was the RNAseq performed on day 7? All previous experiments were performed on day 5. The authors didn't analyze the effect during the early state and cytokines expression. There is no analysis of pro-inflammatory macrophages, specifically F4/80+CD80+. The results do not support their conclusion about significantly attenuating re-epithelization, so I strongly recommend additional experiments. It would be nice to compare the wound closure between WT and the Mrc1-DT group. I am having difficulty comprehending which specific sections of the wound are represented in the images provided at high magnifications.

We sincerely apologize for the significant error regarding the timing of our scRNA-seq analysis. To clarify, we actually performed the scRNA-seq analysis using samples obtained on day 5, not day 7 as incorrectly stated in our original manuscript. This inconsistency likely caused confusion in the interpretation of our results, particularly in relation to our other experiments which were also conducted on day 5. We greatly appreciate the reviewer's careful reading that identified this critical error, which has now been corrected throughout the manuscript.

We added results of further characterization of macrophage subpopulations. *Cd80* levels were high in clusters 1 (*VcanI*⁺) and 3 (*Cd9*⁺), which expressed lower levels of *Adgre1* (F4/80). These *Cd80*^{hi}*Adgre1*^{int} cells were more abundant in *Mrc1-DTR* as compared to control mice (Fig.

3G). As shown in EV Figure 3, these subpopulations expressed the genes related to inflammation and immune response.

We also added images 11 days after wounding in Fig. 1F. As shown in the updated Figure 1F, the wound closure was significantly delayed in *MRC1-DTR* group (Fig. 1F and G) that support our conclusions the delayed re-epithelialization by CD206⁺ cell depletion. For the high resolution images, the areas that were enlarged are shown on the low magnification images (Fig. 5B and 5D).

-Fig 4. To provide more robust support for the conclusion, I recommend conducting additional experiments, including co-staining with markers for different layers. It would be nice to add the representative Trichrome image of the wound. I am having difficulty comprehending which specific sections of the wound are represented in the pictures provided at high magnifications.

We have conducted additional experiments to include co-staining with markers for different layers of the wound (Fig. 5A). The localization of EN-positive fibroblasts was confirmed using CD26 as an upper dermis (papillary dermis) fibroblast marker and FAP as a marker in the lower dermis (reticular dermis) layer. In uninjured skin, EN-positive fibroblasts were scattered throughout the dermal layer (Rinkevich et al. *Science* 348:2151, 2015), whereas in the wound bed, and they localized to the deep dermal layer where CD206⁺ macrophages are accumulated (Fig. 5A and 5B). We have added representative Masson's trichrome-staining images of the wound (Fig. 5F) that illustrate collagenous connective tissue fiber deposition was attenuated in the *Mrc1-DTR* wounds that supporting our findings.

As described in the response to the previous comment, we have added squares to the low magnification views to indicate the region the high magnification views show (Figures 5A, 5B, 5D and 6D).

-Fig 5. I struggle to discern the notable differences between groups; the PDGF-A signal appears unspecific in the immunostaining. To provide more robust support for the effect of

the PDGF-AA treatment, I recommend conducting additional experiments. Again, I struggle to discern the notable differences between groups (Fig 5G.)

We have replaced the PDGF-A staining to better show expression of PDGF-A (Fig. 6B corresponding to the previous version of Fig. 5B). We have also replaced the Ki67/En1 staining images with those better representing the staining (Fig. 6I corresponding to the previous version of Fig. 5G). We apologize for the off-focus images in the previous version of figures.

****Results and Discussion**:** The findings must be interpreted and discussed more in-depth concerning the research question and previous antecedents. Additionally, the authors could further enhance the paper's impact by discussing the implications of their findings beyond Mrc1-DTR transgenic mice and exploring potential avenues for future research in this area. The paper doesn't acknowledge the study's limitations and possible areas for further research.

I recommend that the authors conduct a comprehensive revision of their manuscript to enhance its clarity and depth before considering resubmission for publication.

As suggested, we have added a discussion of the wider implications of our findings and limitations. We have also made it clear where further research is needed. We have also highlighted subpopulation-specific roles in wound healing. Understanding the dynamic communication between macrophages and fibroblasts is important not only for elucidating cellular responses and mechanisms to wounds, but also for therapeutic strategies to improve wound healing (p. 29).

We have expanded the last paragraph of the discussion to emphasize the importance of studying cellular communities in wound healing not only for skin wound healing but also for broader diseases, including age-related diseases (p. 29).

We have also clearly stated the limitations of our study (p. 28). These limitations include the modest effect of CD206+ macrophage depletion, potential effects on other CD206-expressing cell types, and technical limitations of our experimental approach, as highlighted by the reviewers.

Dear Dr. Oishi

Thank you for the submission of your revised manuscript to our editorial offices. I have now received the reports from the three referees that were asked to re-evaluate the study, you will find below. As you will see, the referees now support its publication in EMBO reports. However, referee #2 has remaining concerns and suggestions to improve the manuscript, I ask you to address in a final revised manuscript. Please do the requested changes to the manuscript. Please also provide a final p-b-p-response regarding these points.

- Please provide the abstract written in present tense throughout.
- Please provide an institutional e-mail address for the corresponding author on the title page of the manuscript and in the submission system.
- Please provide individual production quality figure files as .eps, .tif, .jpg (one file per figure), of main figures and EV figures. Please upload these as separate, individual files upon re-submission.
- Please fuse the present EV and Appendix figures to have 5 final EV figures. Please follow the nomenclature Figure EV1, Figure EV2 etc. The figure legend for these should be included in the main manuscript document file in a section called Expanded View Figure Legends after the main Figure Legends section. Finally, please delete the combined Appendix/EV-Figure file.
- Please move all the methods information and related references to the main manuscript text. We do not allow supplementary methods. Then please delete the file with the extra methods and legends and EV tables (see below).
- Tables EV1 and EV2 are datasets. Please upload these as dataset files with a legend on the first TAB named Dataset EV1 and Dataset EV1. Please update all the callouts for this item.
- Appendix Figure S1 contains only tables. Please include these as one table in the main manuscript file, or upload as EV table (named Table EV1) or as dataset (see above). Please update the callouts.
- Please move the ethics section to the methods.
- We now use CRediT to specify the contributions of each author in the journal submission system. CRediT replaces the author contribution section. Please use the free text box to provide more detailed descriptions and do NOT provide your final manuscript text file with an author contributions section. See also our guide to authors: <https://www.embopress.org/page/journal/14693178/authorguide#authorshipguidelines>
- Please order the manuscript sections like this, using these names:
Title page - Abstract - Keywords - Introduction - Results - Discussion - Methods - Data availability section - Acknowledgements (including the funding information) - Disclosure and Competing Interests Statement - References - Figure legends - Expanded View Figure legends
- Please name the section 'Resource availability' 'Data availability section' and remove the referee token. Please make sure the data is public upon publication of the manuscript.
- Please provide for the re-submission a complete author checklist, which you can download from our author guidelines (<https://www.embopress.org/page/journal/14693178/authorguide>). Please insert page numbers in the checklist to indicate where the requested information can be found in the manuscript. The completed author checklist will also be part of the RPF.

- Please check again that the number "n" for how many independent experiments were performed, their nature (biological versus technical replicates), the bars and error bars (e.g. SEM, SD) and the test used to calculate p-values is indicated in the respective figure legends. Please also check that all the p-values are explained in the legend, and that these fit to those shown in the figure. Please provide statistical testing where applicable. Please avoid the phrase 'independent experiment', but clearly state if these were biological or technical replicates. Please also indicate (e.g. with n.s.) if testing was performed, but the differences are not significant. In case n=2, please show the data as separate datapoints without error bars and statistics. See also:

<http://www.embopress.org/page/journal/14693178/authorguide#statisticalanalysis>

If $n < 5$, please show single datapoints for diagrams. Moreover:

- Please note that the exact p values are not provided in the legends of figures 1C, E, G, I, L; 5C, E; 6C, E, F, H, J
 - Please indicate what */ **/ ***/ **** represents; if this represents p value(s), please indicate the statistical test used and where appropriate, specify the exact p value in the legend(s) of figure(s) EV2 A-C
 - Please indicate the statistical test used for data analysis in the legend of figure 5C
 - Please note that information related to n is missing in the legends of figures 3D, 4F; EV2 A-C
 - Please note that the error bars are not defined in the legends of figures 5C, E; 6F, J; EV2 A-C
 - Please note that the white arrow heads are not defined in the legend of figure 5D, 6I. This needs to be rectified.
- Please add to each legend (main and EV figures, where applicable) a 'Data Information' section (or name the provided section like this) explaining the statistics used or providing information regarding replicates and scales. See:

- Please add scale bars of similar style and thickness to microscopic images, using clearly visible black or white bars (depending on the background). Please place these in the lower right corner of the images themselves. Please do not write on or near the bars in the image but define the size in the respective figure legend. Presently, some scale bars are too thin or have text nearby. Please check.

- Please make sure that all the funding information is also entered into the online submission system and that it is complete and similar to the one in the acknowledgement section of the manuscript text file. Presently, the Takeda Science Foundation is missing in the submission system. Please check.

- Please make sure that all figure panels (main and EV figures) are called out separately and sequentially. Presently, there is no callout for Figure 2H. Please check.

- All Materials and Methods need to be described in the main text using our 'Structured Methods' format, which is required for all research articles. According to this format, the Methods section should include a Reagents and Tools Table (listing key reagents, experimental models, software, and relevant equipment and including their sources and relevant identifiers), uploaded as separate file, and a Methods section in which we encourage the authors to describe their methods using a step-by-step protocol format with bullet points, to facilitate the adoption of the methodologies across labs. More information on how to adhere to this format as well as downloadable templates (.doc) for the Reagents and Tools Table can be found in our author guidelines (section 'Structured Methods'):

- The left diagram of Fig. 1B is also shown in figure EV1A. Moreover, the diagrams in each column of Fig. EV1B seem similar. I suppose this is intentional. Please mention and explain the reuse in more detail in the respective legends.

In addition, I would need from you uploaded separately:

Please let me know if you have questions regarding the revision.

Best,

Referee #1:

Although the authors were unable to develop novel mechanistic insights, the modifications to the text including limitations of the study are reasonable.

Referee #2:

The strength of the manuscript is a detailed single-cell analysis of the impact of deleting CD206+ cells on cell populations, particularly fibroblasts. This appears to be well done. However, the underlying hypothesis that CD206+ macrophages communicate with fibroblasts through PDGFA to enhance wound healing is not well established by the data presented. Thus, there may be important data, but it should be re-cast as a single-cell analysis (supported by other approaches) of the cellular impact of deleting M2 macrophages, particularly on fibroblasts, rather than on a key signaling pathway.

Major comments.

1. It is well known that CD206+ cells and PDGFA are key factors in wound healing. However, the underlying hypothesis involving a specific communication pathway is substantially undercut by a very modest impact of deleting CD206+ cells. The simple interpretation of the data is that there was a technical limitation in the approach based on previous studies with M2 macrophages. In contrast, the data showing an impact of M2 macrophage depletion on various fibroblast populations appears more convincing.
2. A limitation is that the scRNAseq data does not point to CD206+ cells as the predominant source of PDGFA. This could be shown in a UMAP of PDGFA in the initial clustering or through a FindAllMarkers table in R/Seraut, but I could not find this data. The IF data to make this claim is difficult to interpret since a limited field was shown.
3. The FMO suggests that there is a background problem in Fig 1C with the CD206 antibody. In the experimental figure Q2 as 8.57% of cells and there is 1.68% in Q2 in the FMO. Thus, 20% of the signal detected is background, which is high.

Referee #3:

The authors have adequately addressed the reviewers' comments in the revised version of the manuscript. I recommend publishing in EMBO Reports.

Response to Referee #2:

We thank the Referee for the comments regarding our manuscript. We greatly appreciate the effort and time taken to provide us with a number of constructive comments to improve this article. (Excerpts from the reviewer's critique are indicated in bold.)

The strength of the manuscript is a detailed single-cell analysis of the impact of deleting CD206⁺ cells on cell populations, particularly fibroblasts. This appears to be well done. However, the underlying hypothesis that CD206⁺ macrophages communicate with fibroblasts through PDGFA to enhance wound healing is not well established by the data presented. Thus, there may be important data, but it should be re-cast as a single-cell analysis (supported by other approaches) of the cellular impact of deleting M2 macrophages, particularly on fibroblasts, rather than on a key signaling pathway.

Major comments.

1. It is well known that CD206⁺ cells and PDGFA are key factors in wound healing. However, the underlying hypothesis involving a specific communication pathway is substantially undercut by a very modest impact of deleting CD206⁺ cells. The simple interpretation of the data is that there was a technical limitation in the approach based on previous studies with M2 macrophages. In contrast, the data showing an impact of M2 macrophage depletion on various fibroblast populations appears more convincing.

We sincerely appreciate the reviewer's valuable and insightful comments. We acknowledge that the modest effect of CD206⁺ cell depletion, potentially attributable to the relatively high number of CD206⁺ cells that remained after DT treatment, is a limitation of our study. As the reviewer correctly pointed out, our scRNA-seq analysis and immunostaining clearly demonstrated that CD206⁺ cell depletion significantly altered fibroblast subpopulations during wound healing. While we identified PDGFA signaling as one of the intercellular communication pathways between CD206⁺ macrophages and Gpnmb^{hi} fibroblast subsets, our data suggest that multiple pathways are likely involved in this complex interaction (Figure EV3). We have clearly stated these points in the Discussion (p. 27). Following the reviewer's suggestion, we have modified abstract to emphasize the broader role of CD206⁺ macrophages in modulating fibroblast heterogeneity and function during wound healing, rather than focusing

exclusively on a single signaling pathway (p.2).

2. A limitation is that the scRNAseq data does not point to CD206+ cells as the predominant source of PDGFA. This could be shown in a UMAP of PDGFA in the initial clustering or through a FindAllMarkers table in R/Seraut, but I could not find this data. The IF data to make this claim is difficult to interpret since a limited field was shown.

We greatly appreciate your pointing out this important limitation. As suggested, we have added a UMAP visualization of *Pdgfa* expression across all celltypes in Fig. 2D. Results indicate that the *Pdgfa* mRNA is expressed in macrophage population (in particular, the *Mrc1*⁺ macrophage population) although *Pdgfa* mRNA is expressed in other cell types such as Langerhans cells, endothelial cells and epithelial cells. This information is also added in the main text (p. 20).

3. The FMO suggests that there is a background problem in Fig 1C with the CD206 antibody. In the experimental figure Q2 as 8.57% of cells and there is 1.68% in Q2 in the FMO. Thus, 20% of the signal detected is background, which is high.

We appreciate the reviewer's insightful observation regarding the CD206 antibody background. Our FMO results demonstrate clear distinction between stained and unstained cell populations. However, the FMO control shows 1.68% cells in Q2, while the experimental sample shows 8.57%, indicating some background signal. This level of background (approximately 20% of detected signal) is consistent with the technical limitations inherent to flow cytometry when analyzing cells with high autofluorescence, such as macrophages. We performed additional analysis focusing on CD206^{hi} populations, which exhibited lower background signal (Figure EV1C). This reanalysis showed that the CD206^{hi} population was substantially reduced in *Mrc1-DTR* mice (Fig. 1C), further strengthening the main conclusions of our study regarding the specific depletion of CD206^{hi} macrophages in our model system.

Dr. Yumiko Oishi
Institute of Science Tokyo
1-5-45, Yushima
Bunkyo, Tokyo 1138510
Japan

Dear Dr. Oishi,

Thank you for the submission of your further revised manuscript. As you know, I have received a rather critical report from the referee that was asked to re-evaluate the further revised manuscript, which I include below again. I also received your rebuttal letter regarding the report of the referee.

Going through your letter and the revised manuscript, also considering that the two other referees have been satisfied by the revisions and support publication, I think the manuscript has been sufficiently revised.

I am thus very pleased to accept your manuscript for publication in the next available issue of EMBO reports. Thank you for your contribution to our journal.

Yours sincerely,

Referee #2:

My primary comment on the initial submission was that the manuscript had important data but should be re-cast as a single-cell analysis (supported by other approaches) of the cellular impact of deleting M2 macrophages, particularly on fibroblasts, rather than on a key signaling pathway. This was due to the very modest effect of CD206+ cell depletion on healing, which was at odds with the majority of previous reports. Thus, there was very little to "rescue" and the emphasis on PDGF-AA rescue was misleading. However, an appropriate response to re-focus on the fibroblast phenotype, function and multiple communication pathways would require a new bioinformatic analysis that was not done and a revision of the paper with a new emphasis, that was not done.

Major Comment: While the authors now recognize in the response to reviewer comments that the data do not support the emphasis on PDGFAA signaling, they fail to make substantial changes in the manuscript. The Abstract, Results and Discussion do not fully emphasize the impact that CD206+ cells have on fibroblasts and resort to focusing on PDGF-AA rescue, which is very modest as mentioned in my previous review. Little mention is made of the impact of deleted CD206+ on fibroblast phenotype, function or multiple communication pathways. This deficit occurs throughout the manuscript. Most importantly, virtually no new bioinformatic analysis has been added to bring for these points. The lack of responsiveness of the authors is disappointing.